# A Late Cretaceous true polar wander oscillation

Ross N. Mitchell [1,2✉], Christopher J. Thissen [3], David A. D. Evans [3], Sarah P. Slotznick [4], Rodolfo Coccioni [5], Toshitsugu Yamazaki [6] & Joseph L. Kirschvink [2,7]

True polar wander (TPW), or planetary reorientation, is well documented for other planets and moons and for Earth at present day with satellites, but testing its prevalence in Earth's past is complicated by simultaneous motions due to plate tectonics. Debate has surrounded the existence of Late Cretaceous TPW ca. 84 million years ago (Ma). Classic palaeomagnetic data from the Scaglia Rossa limestone of Italy are the primary argument against the existence of ca. 84 Ma TPW. Here we present a new high-resolution palaeomagnetic record from two overlapping stratigraphic sections in Italy that provides evidence for a ~12° TPW oscillation from 86 to 78 Ma. This observation represents the most recent large-scale TPW documented and challenges the notion that the spin axis has been largely stable over the past 100 million years.

[1] State Key Laboratory of Lithospheric Evolution, Institute of Geology and Geophysics, Chinese Academy of Sciences, Beijing, China. [2] Division of Geological and Planetary Sciences, California Institute of Technology, Pasadena, CA, USA. [3] Department of Geology and Geophysics, Yale University, New Haven, CT, USA. [4] Department of Earth Sciences, Dartmouth College, Hanover, NH, USA. [5] Honorary Professor, Università di Urbino, Urbino, Italy. [6] Atmosphere and Ocean Research Institute, The University of Tokyo, Chiba, Japan. [7] Earth-Life Science Institute, Tokyo Institute of Technology, Tokyo, Japan. ✉email: ross.mitchell@mail.iggcas.ac.cn

True polar wander (TPW) is the reorientation of a planet or moon in order to align the body's greatest nonhydrostatic principal axis of inertia ($I_{max}$) with the spin axis[1–4]. On Earth, TPW is achieved by wholesale rotation of the solid, silicate Earth (mantle and crust) around the liquid outer core. As Earth's magnetic pole is tied primarily to rotationally induced excitations of the outer core, the magnetic poles remain aligned with the rotation axis through a TPW event. Thus, palaeomagnetic data record TPW as the coherent, simultaneous motion of all coeval rocks about a single equatorial Euler pole defined by Earth's minimum moment of inertia ($I_{min}$). Palaeomagnetic sampling of a continuous sedimentary succession is an effective single-locality test of TPW as it eliminates potential caveats such as differential remagnetization and tectonic structure[5,6].

The possibility of Late Cretaceous TPW is hotly debated[7–10]. One TPW rotation has been proposed to occur between the emplacement of two seamounts with Ar-Ar ages overlapping at ca. 84 Ma and palaeomagnetic poles discordant by $16° ± 3°$ (refs. [9,10]). It has been asserted, however, that classic palaeomagnetic data from the correlative Gubbio and Moria sections of Scaglia Rossa limestone do not permit ca. 84 Ma TPW[7]. Irrespective of the quality of the original data, an analysis that calculates only three average inclinations from 90 to 75 Ma[7] does not constitute a robust test of a comparatively rapid[11] process. But the reliability of those old (however seminal) data[12,13] is questionable for directional studies[8]. In particular, the Gubbio and Moria data pre-date least-squares analysis of palaeomagnetic data[14], the use of controlled-atmosphere thermal demagnetization, and the measurement by sensitive superconducting magnetometers.

In this work, we present >1000 palaeomagnetic data from the Scaglia Rossa limestone as a rigorous test of the ca. 84 Ma TPW event. Samples were collected from two parallel stratigraphic sections as a test of reproducibility. Modern demagnetization and analytical palaeomagnetic methods were employed, including state-of-the-art rock magnetic experiments that shine new light on the origin of the stable palaeomagnetic remanences of the Scaglia Rossa limestone. Both stratigraphic sections definitively confirm the existence of ca. 84 Ma TPW. Furthermore, our new high-resolution record suggests not only a single TPW shift at this time, but a "roundtrip" TPW oscillation where the pole excurses and then returns back to its original pole position.

## Results

**Stratigraphic sections**. Samples were collected from stratigraphically correlative sections at Apiro and Furlo in Italy (Supplementary Fig. 1) at typical stratigraphic intervals of 5, 10, 25, 50, or 100 cm (Supplementary Fig. 2). At Apiro, the R1 and R2 members of the Scaglia Rossa are devoid of slumps and are lithologically homogenous apart from radiolarian chert beds in the R1 member. Only the homogenous background lithology of pelagic limestone was sampled for palaeomagnetic analysis. At Furlo, sampling was limited to strata in between slumping events known to anomalously rotate declination due to rotation about a local vertical axis[15]. Furthermore, stylolitic "pseudo-bedding" that characterizes the bedding style of the Scaglia Rossa at Gubbio and elsewhere in the Umbria–Marche basin[16] is absent at Apiro, and less pronounced at Furlo (Supplementary Fig. 2), suggesting the sedimentary rocks of our sections are less tectonically modified than others.

**Magnetic mineralogy**. We used traditional palaeomagnetic methods for analysis on an automated system[17] and employed a detailed thermal demagnetization scheme (conducted in a controlled nitrogen atmosphere) to facilitate separation of magnetic components (Methods). The reliability and quality of Scaglia Rossa palaeomagnetism is well known[12,13,15,16]. Previous studies on similar sedimentary rocks generally report stable behavior up to ~570 °C, indicative of magnetite being the predominant remanence carrier despite the red color of the Scaglia Rossa imparted by fine-grained authigenic hematite[18]. First-order reversal curve (FORC) distributions are dominated by the "central ridge" (Fig. 1c), which indicates negligible magnetostatic interaction among magnetic grains and is interpreted to result from a chain structure of bacterially formed magnetite[19]. Additional rock magnetic experiments indicating a similar magnetic mineralogy for all lithologies and magnetization directions are presented in Supplementary Figs. 3 and 4. The presence and dominance of magnetofossils in the Scaglia Rossa limestone can account for its exceptional palaeomagnetic stability (Fig. 1a, b), making it an ideal candidate for testing ca. 84 Ma TPW.

**Demagnetization**. Virtually all Scaglia Rossa samples carry low-stability magnetic overprints coincident with the present local field, consistently removed after thermal demagnetization to 150 °

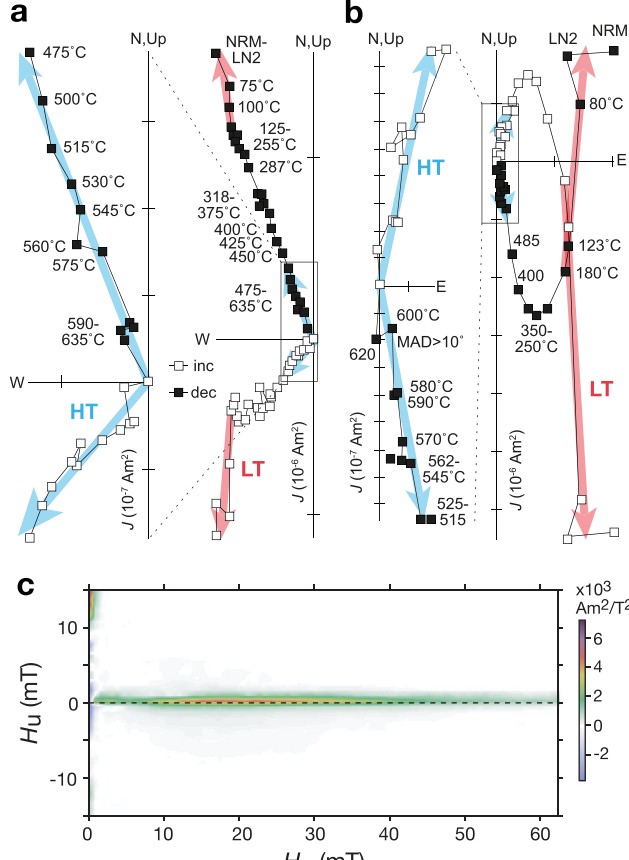

**Fig. 1 Magnetization of the Scaglia Rossa limestone. a**, **b** Orthogonal vector diagrams of typical demagnetization behavior with low-temperature (LT) components and insets with magnified views of high-temperature (HT) components. NRM natural remanent magnetization, LN2 liquid nitrogen immersion. **a** Normal polarity sample (FUR2-136) from Furlo section. Steps after significant loss in magnetization >575 °C were excluded from least-squares fit. **b** Reversed polarity sample (C10AD34) from Apiro section. Steps >590 °C with circular standard deviation (CSD) >10° were excluded from least-squares fit. **c** First-order reversal curve (FORC) distribution exhibiting a "central ridge" indicating the magnetic mineral assemblage of the Scaglia Rossa limestone is dominated by highly stable, single-domain, biogenic magnetite magnetofossils. San Severino locality, near the main Apiro section. See Methods for experiment details.

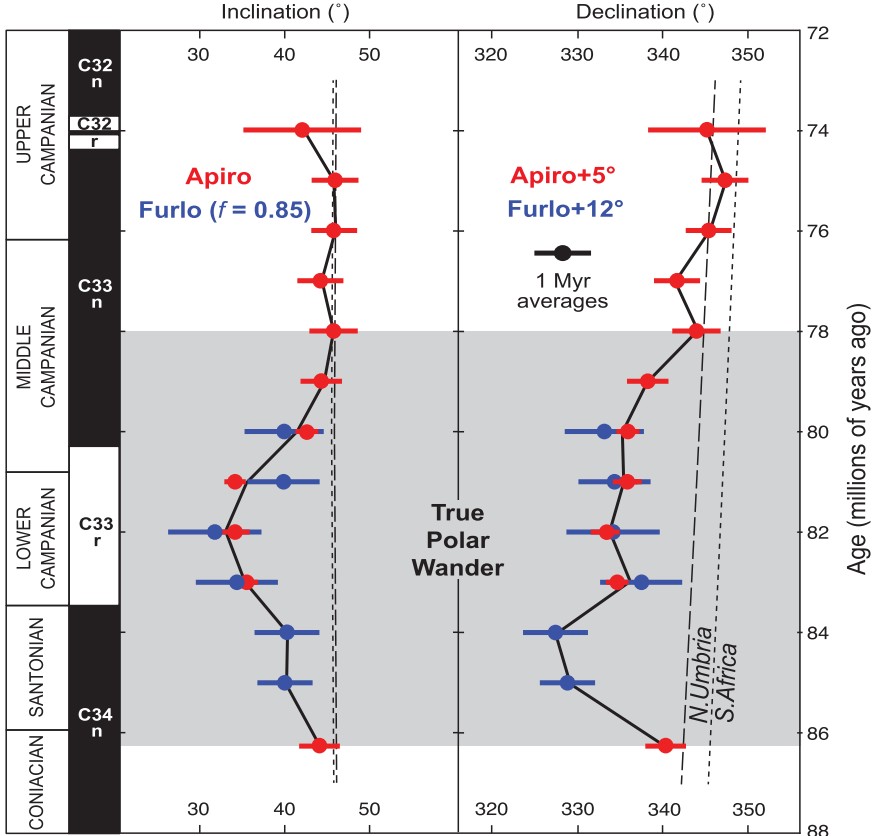

**Fig. 2 Palaeomagnetic inclination and declination of Italy in Late Cretaceous time.** One-million-year averages and error bars of the 1090 samples were calculated using Fisher statistics (excluding outliers interpreted as geomagnetic excursions, spurious magnetizations, or slumps; reversed polarity data were reversed for mean calculations). The temporary ~10–15° excursion of both inclination and declination is interpreted as a true polar wander oscillation. Directional predictions for a representative site (43.5°N, 12.9°E) from the global apparent polar wander path of Torsvik et al.[72] and the tectonic model of van Hinsbergen et al.[71] are shown in both North Umbrian and South African coordinates. Apiro declination data have been adjusted for local rotation to match those reference frames (+5°), and Furlo data have been corrected to match optimally the Apiro dataset with local rotation (+12°) and inclination shallowing[75] (f = 0.85). Global polarity time scale[76] shown on left. Age models provided in the Methods. Supplementary Fig. 7 and Supplementary Data 1 for raw data before averaging.

C, matching remanence expectations for either secondary goethite or a viscous overprint carried by large magnetite grains (Supplementary Fig. 5 and Supplementary Data 2). After removal of the overprint, the characteristic magnetizations were then typically stable up to 500–580 °C indicative of magnetite. All characteristic remanent magnetizations were determined by linear decay to the origin or stable-end point behavior on orthogonal demagnetization diagrams and exhibit striking palaeomagnetic stability for sedimentary rocks (Fig. 1a, b). The isolation of primary Late Cretaceous remanences is collectively supported by the successful removal of present local field overprints up to 150–180 °C (Fig. 1a, b and Supplementary Fig. 5), magnetostratigraphy matching the geologic time scale (Fig. 2 and Supplementary Figs. 6 and 7), and the success of fold tests conducted in the area[13].

**Palaeomagnetic directions**. Exploring the possibility of long-term variations in palaeomagnetic direction, we calculate the average inclination and declination of Italy in one-million-year intervals (Fig. 2) and its associated palaeopole path (Fig. 3). Palaeomagnetic data from Italy must be locally restored relative to the larger African plate, which also must be tectonically reconstructed in order to test the TPW hypothesis (Methods). Contrary to the claim that the Scaglia Rossa limestone does not show significant variations in inclination[7], our new data show a ~12° oscillation (~24° total) between 86 and 79 Ma (~7 million

years [Myr]; Fig. 3) that temporarily both rotated and translated Italy to lower latitudes during C33r, with coeval changes to both declination and inclination, respectively (Fig. 2). Although the excursion broadly overlaps with magnetochron C33r, we note that the peak offset occurs during C34n, arguing against the idea that magnetic polarity contaminates the signal. Furthermore, our demagnetization methods are detailed enough to resolve and remove overprints (Fig. 1a, b and Supplementary Fig. 5) and directional transitions observed across magnetic reversals are smooth instead of sudden (Fig. 2), as would be the case if directional shifts were artifacts of unresolved normal polarity overprints.

**Inclination shallowing**. Inclination shallowing due to sediment compaction is a common problem in sedimentary palaeomagnetism that must be considered to explain the temporary lowering of inclination. Inclination shallowing usually stems from the electrostatic attraction of small magnetite lathes to clay flocs or hematite platelets having a preferred flow-regime or gravity-induced directionality[20]. But with the Scaglia Rossa, we are not relying on nanoscale magnetite laths or on hematite platelets for our magnetization, rather, single-domain biogenic magnetite grains (Fig. 1c). The Scaglia Rossa magnetofossils may well be independent of the clay associated or grain shape anisotropic effects that commonly cause inclination shallowing. Nonetheless, to test for burial compaction, we conducted anisotropy of

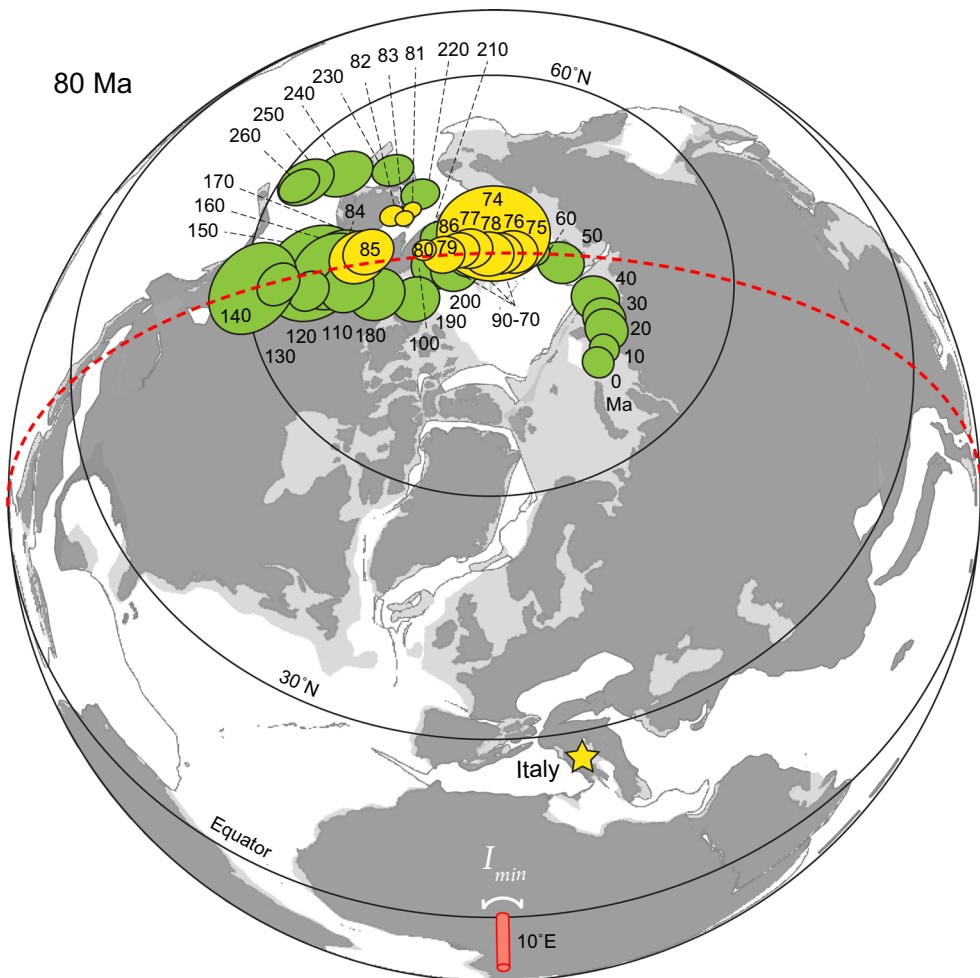

**Fig. 3 Mesozoic true polar wander.** Reconstruction at 80 Ma, centered at 10°E palaeolongitude, showing globally averaged palaeopoles[72] (green) exhibiting oscillations interpreted as earlier Mesozoic TPW events[22]. Italy and its new poles (yellow) reconstructed using Euler parameters (4.0°N, 32.6° W, 21.8°CW) in a global plate model with simplified Mediterranean motions guided by ref. [71]. All Italian poles are from Apiro except 85 and 84 Ma from Furlo that fill the gap at Apiro (Supplementary Fig. 2). Error ellipses are projections of cones of 95% confidence interval. Present-day TPW axis (minimum moment of inertia, $I_{min}$) at 0°N, 10°E and orthogonal plane of TPW in red[4, 25]. Note consistency of TPW longitude in all datasets. Supplementary Data 3 for palaeopoles.

magnetic susceptibility (AMS) experiments. Previous AMS data at Apiro and Furlo reveal a minor clustering of $k_{min}$ axes and girdle of $k_{max}$ and $k_{int}$ axes, implying a mildly oblate compaction fabric in the plane of bedding[21]. Our new data from the same sections confirm a minor oblate flattening in the plane of bedding generally for all samples. However, no correlation between AMS and inclination is found when the data are cross-plotted (Supplementary Fig. 8), suggesting inclination shallowing is unlikely to explain the inclination variation within a section. Finally, the distinct data groupings through time (Fig. 2) differ substantially in declination, not just inclination, so inclination shallowing cannot entirely account for the systematic dispersion of the Scaglia Rossa data.

## Discussion

As predicted by the ca. 84 Ma TPW hypothesis[9], the largest polar excursion observed in Italy occurs between 84 and 82 Ma (Fig. 3). Our Italy data thus do not refute the ca. 84 Ma TPW hypothesis[9]. The amplitude of the ~12° ± 3° excursion of Italy palaeopoles (Fig. 3) furthermore overlaps within uncertainty the 16° ± 3° dispersion of Pacific Plate palaeopoles[9,10]. TPW is able to explain the coincident excursions in both inclination and declination observed in Italy (Fig. 2). Due to the palaeogeographic position of

Italy relative to the TPW axis near Africa[22], Italy should have experienced both latitude change (inclination) as well as rotation (declination) during the TPW excursion, as is observed. The TPW axis ($I_{min}$) is defined by the triaxial or nearly prolate shape of the nonhydrostatic Earth due to long-wavelength mantle convection[23,24], which is thought to be stable over the past 300 Myr during the Pangaea supercontinent cycle[22]. The east-west longitudinal orientation of the polar motion observed in the Late Cretaceous in Italy is similar to that of earlier Mesozoic TPW[22] as well as the present-day TPW axis[4,25], as expected (Fig. 3). Also, as observed elsewhere[5] and in high-resolution detail here (Figs. 2 and 3), TPW is typically modeled as a back-and-forth "roundtrip" oscillation where the pole shifts away, but then snaps backs to the original pole position[23]. Whether the "figure 8" path exhibited by both Mesozoic TPW oscillations (Fig. 3) is significant remains to be tested with modeling.

A cumulative "roundtrip" amplitude of ~24° for the TPW oscillation observed in Italy over ~8 Myr (86–78 Ma) implies a TPW rate of ~3.0° Myr$^{-1}$. The present viscosity of the lower mantle sets a ~2.4° Myr$^{-1}$ "speed limit" for TPW as the solid Earth must conform to the migrating hydrostatic bulge for reorientation to proceed[11]. Our data are thus broadly compatible with this theoretical speed limit, but may suggest that the

Cretaceous mantle was different than today. As detectable secular mantle cooling has occurred since Late Cretaceous time[26], the TPW speed limit could have been higher in the past with a hotter, less viscous mantle[11]. Despite consistency between the Late Cretaceous petrological[26] and palaeomagnetic data, the large uncertainties in the parameters of the mantle properties from which the TPW speed limit is determined[11] renders the implication for a hotter mantle at this time as constrained by TPW rate ambiguous at best. Instead, older, and potentially faster TPW events[24], should be used to constrain such a problem. The slightly above average speed of ca. 84 Ma TPW could rather be explained by a larger mantle convective forcing than normal.

Our observed TPW oscillation of ~12° amplitude at this age can be used to parameterize models to better assess Earth's nonhydrostatic figure and the relative effects of sinking slabs and rising mantle plumes on Earth's rotational stability. Using the well-constrained history of recently subducted slabs, the modeled effect on Earth's nonhydrostatic geoid that controls the location of the spin axis (and thus TPW) predicts a large and rapid TPW event between 80 and 90 Ma[27], precisely when our observations indicate that TPW in fact occurred (Figs. 2 and 3). This general agreement thus begins to reconcile the apparent discrepancy between TPW observations and the model predictions at this age[27]. Nonetheless, despite the striking consistency in suggesting large-scale (>10°) TPW between 80 and 90 Ma, there is a notable contrast in the amplitudes and styles of the observed and predicted TPW events. Our data suggest a ~12° TPW oscillation, whereas the model[27] predicts a ~90° TPW shift. Since the model considers only TPW excitation from subduction[27], we speculate that plumes rising through the mantle at this time may have counteracted the effect of sinking slabs and thereby reduced the amplitude of TPW excited by subduction alone as plumes have the opposite effect of slabs on Earth's geoid kernel[27]. Plumes leaving the core-mantle boundary (e.g., the Réunion hotspot later producing the ca. 66 Ma Deccan Traps) or those reaching the surface (e.g., the Marion hotspot producing the 84–91 Ma Morondova large igneous province) at this age, both of deep mantle origins and occurring in the Indian Ocean of the southern hemisphere[28], would have been well positioned to have potentially counteracted the effect of subducting slabs largely in the North Pacific[27]. Such complications aside, on both a theoretical and empirical basis, Late Cretaceous TPW represents the youngest known large-scale TPW event in Earth history[24].

TPW predicts that all continents rotated at this age and about a common TPW axis consistent with palaeogeography. Although the Pacific Plate exhibits a polar shift of similar age and magnitude to Italy[9,10], uncertainty of the palaeogeographic reconstruction of the Pacific Plate precludes testing the consistency of the palaeolongitude of the TPW axis. Palaeopoles of this age from Europe and North America, continents that are both well resolved palaeogeographically with restoration of Atlantic seafloor spreading, both circumscribe polar excursions of similar magnitude as well as similar coplanar orientation to Italy, preliminarily passing the global reproducibility test of the TPW hypothesis (Supplementary Fig. 9; ref. [9]). Unlike earlier Mesozoic TPW events that are substantiated by the palaeomagnetic records of multiple or even most continents[22,29–32], Late Cretaceous TPW is nonetheless only documented for Italy, the Pacific Plate, and, with less confidence, a few other regions. The relative paucity of sampling is not surprising, however, provided the rapidity of the proposed event. Future testing of coeval stratigraphies around the world should be conducted at commensurate resolution on palaeomagnetic targets of comparable quality. It should also be noted that the older portion of the TPW oscillation depicted here has only been documented in Italy and thus requires additional testing in particular.

Even though this time interval has been examined by other palaeomagnetic studies, their sampling resolutions were likely insufficient to identify the TPW oscillation and thus aliased it. Our study collected an order of magnitude more samples than is typically done in high-quality palaeomagnetic studies over a 10 Myr interval, i.e., ~1000 samples instead of ~100. Despite arguing for multiple TPW events between 300 and 100 Ma (Fig. 3), global appraisals of TPW have not identified any TPW events in the past 100 Myr[22,33]. However, such analyses employ both 10-Myr averaging of palaeomagnetic data and use data from multiple continents to generate global apparent polar wander (APW) paths. Indeed, TPW should be a global phenomenon, which is why such an approach is taken. Nonetheless, while this method may be adequate for identifying larger amplitude TPW shifts (20–30°) occurring over longer amounts of time (≥10 Myr), it would smooth out and/or obscure the signal of a smaller amplitude TPW shift (~10°) over a shorter amount of time (≤10 Myr). By contrast, as allowed by our increased sampling resolution, our study employs 1-Myr averaging of data from one continent. Although TPW for the past 100 Myr may be of smaller amplitude[34,35] than earlier TPW shifts[24], our high-resolution palaeomagnetic study would suggest that one cannot rule out the possibility of rapid TPW shifts of small amplitude (<20°) in the most recent past.

## Methods

**Geologic setting and sampling.** The Umbria–Marche pelagic succession of central Italy is the classic locality for the Cretaceous-Palaeogene (K/Pg) extinction[36] and several of the oceanic anoxic events, where they occur within rhythmically bedded limestones, marls, and cherts[37,38]. This study focuses on a ~15-Myr interval between these dramatic Late Cretaceous events. Much previous work has been done to date the Umbria–Marche succession. The Gubbio section (Supplementary Fig. 1) spans all of Late Cretaceous time and serves as a standard for magnetostratigraphic sampling[12]. Although relatively stratigraphically condensed, a subsequently sampled magnetostratigraphy at Moria (Supplementary Fig. 1) confirms the magnetic reversal pattern observed at Gubbio and on the seafloor, as well as serving as a positive palaeomagnetic fold test validating the Scaglia Rossa palaeopole[13].

Both the classic Gubbio and Moria sections, unfortunately, are tectonically disturbed with faults and slumps, often causing cryptic block rotations vital to avoid[15,39,40], at the base of the R2 Member of the Scaglia Rossa Formation, our target interval. We sampled two correlative stratigraphic sections in distinct thrust panels (Supplementary Figs. 1 and 2). The Apiro section (43.38°N, 13.155°E) spans the entire ~10-Myr-long magnetochron 33 and has been studied previously for rock magnetic fabrics[41]. The Furlo section (43.64°N, 12.71°E) is exposed on the western flank of a large anticline and was previously sampled for magnetostratigraphy[15]. By correlation, the Furlo record fills in a ~2-Myr gap in lower C33r at Apiro. Furlo also offers reproducibility tests with Apiro.

Throughout the Umbria–Marche basin, the Scaglia Rossa was deposited above the carbonate compensation depth, but below storm wave base; estimates of depth range from 1500 to 2000 m (refs. [42,43],). The basin formed on the thinned continental margin of Adria, which was tectonically active as evidenced by synsedimentary slumping, faulting, and turbidity currents[39,44]. The presence of active tectonics has complicated the retrieval of a continuous magnetostratigraphic record during Santonian–Campanian time in the basin: of measured sections of Scaglia Rossa, half exhibit soft sedimentary deformation accommodated by a marly layer occurring around the C33n-C33r reversal throughout the basin. U/Pb constraints from across the Umbria–Marche Apennines[45] allows correlation from the basin (e.g., Gubbio) to the southern platform[46].

The Scaglia Rossa is composed of three principal lithologies (light pink pelagic limestone, nodular or tabular chert of assorted colors, and dark pink marl), sometimes deposited in cycles that are striking at the outcrop scale. Only the homogenous background lithology of pelagic limestone was sampled for palaeomagnetic analysis (Supplementary Fig. 2). The dominant lithology, a white to light pink pelagic biomicrite, is made up of abundant foraminifera tests suspended in a nannofossil (largely coccolith) matrix, with minor calcified, sub-spherical radiolarian skeletons. The partially silicified nodular and tabular chert layers of various colors are instead made of an abundant assemblage of radiolarians suspended in a microcrystalline quartz matrix and few silicified, planktonic foraminifera tests that are often pyritized.

Apiro has never been studied within a stratigraphic context. Red chert layers near the base of the exposure can be lithologically identifiable as the top of the R2 Member of the Scaglia Rossa. Apiro contains all of the Scaglia Rossa Formation, with exception of a ~30-m-thick covered interval near the base of C33r. Apiro is stratigraphically expanded and devoid of stylolitic "pseudo-bedding," which

characterizes the bedding style of the Scaglia Rossa elsewhere in the Umbria–Marche basin[47]. Correlation to the relatively constant thin beds of Furlo reveals much thicker beds and faster sediment accumulation at Apiro. Turbidites and evidence for resedimentation are not found at Apiro as in Furlo. Data from Furlo can be imported into the Apiro record in order to fill in the missing C34n-C33r reversal interval (Supplementary Fig. 2 and Fig. 3). Combined, our composite Italy section spans 4 geomagnetic reversals, 7 biozones, and ~15 Myr (88–73 Ma; see section below for details of age modeling).

**Palaeomagnetism**. Oriented palaeomagnetic cores 2.54 cm in diameter were drilled on site using a portable, gasoline-powered, diamond-tipped coring drill with both air- and water-cooled functionality. All 1090 samples were hand-drilled cores except for 24 block samples. Sun-compass observations were made whenever possible to check the magnetic compass, and to independently measure the local magnetic deviation, which, at ~1°E, is indistinguishable from the International Geomagnetic Reference Field model for the sites. Remanent magnetization measurements were made with a 2G Enterprises™ DC-SQuID magnetometer with background noise sensitivity of $5 \times 10^{-12}$ Am² per axis at both the California Institute of Technology and Yale University. The magnetometers are equipped with computer-controlled, online alternating-field (AF) demagnetization coils and an automated vacuum pick-and-put sample-changing array[17]. Samples and instruments are housed in a magnetically shielded room with residual fields <100 nT throughout the demagnetization procedure and <5 nT in the sense region.

After measurement of natural remanent magnetization (NRM), all samples were demagnetized cryogenically in a low magnetic field-shielded liquid nitrogen bath in an attempt to help unblock larger multi-domain magnetite grains by "zero-field" cycling through the Verwey transition near 77 °K (ref. [48]), and typically this step was repeated a second time. Next, all samples were thermally demagnetized in steps of 4–25 °C, starting at 50 °C and going up to 590 °C (or until thoroughly demagnetized or unstable, usually 20–30 thermal steps per specimen) in a magnetically shielded furnace (±2 °C relative error) in a flowing nitrogen atmosphere. After each demagnetization step, automated three-axis measurements were made in both sample-up and sample-down orientations, and samples with circular standard deviation >10° were rerun manually.

Magnetic components were computed for each sample using principal component analysis[14] as implemented in PaleoMag OS X[49] (Fig. 1a, b). See main text for discussion of results. Palaeomagnetic poles from various previous works are compiled and compared using the software programs GPlates (www.gplates.org) (Fig. 3) and Paleomac[50] (Supplementary Fig. 9). To first order, the records between Apiro and Furlo are consistent with each other as demonstrated in Fig. 2. Nonetheless, due to appreciable uncertainties in the different local tectonic restorations and degrees of inclination shallowing in the two sections (see below), we do not combine the data from the two sections when calculating palaeopoles. Instead, we use exclusively poles from Apiro with the exception of the 85 and 84 Ma poles from Furlo that fill the temporal gap at Apiro (Supplementary Fig. 2 and Supplementary Data 1).

**Rock magnetism**. Non-destructive rock magnetic experiments were performed on seven samples of the Scaglia Rossa limestone (six samples from Apiro and one sample from Furlo) using a 2G Enterprises SQuID magnetometer following the RAPID protocols, and analyzed using the RAPID Matlab scripts[17] (https://sourceforge.net/projects/paleomag/). Our protocol includes measurements of AF demagnetization of the NRM, acquisition and demagnetization, anhysteretic remanent magnetization (ARM) acquisition and demagnetization, isothermal remanent magnetization (IRM) acquisition and demagnetization, and backfield IRM acquisition. These analyses can be used to observe fundamental properties that can be used to distinguish different ferromagnetic minerals. The destructive rock magnetic technique of KappaBridge thermal susceptibility was measured on a neighboring specimen using an AGICO MFK1-FA KappaBridge and reduced in Matlab. AMS measurements were conducted using an AGICO MFK1-FA KappaBridge.

FORC diagrams provide information on the distribution of coercivities ($H_c$) and local interaction fields ($H_u$) for a magnetic grain assemblage in a sample[51,52]. In particular, they are used for detecting biogenic magnetite in sediments[53,54]; the occurrence of a sharp ridge along the $H_c$ axis with little vertical spread, called "the central ridge," is diagnostic, which indicates negligible magnetostatic interaction among magnetic grains and is interpreted to result from a chain structure of bacterial magnetite grains[19,55]. FORC measurements were made on limestone samples from a section at San Severino near Apiro (C33r/C33n reversal age) using an alternating gradient magnetometer (MicroMag 2900, Princeton Measurements Corporation) at the Geological Survey of Japan, National Institute of Advanced Industrial Science and Technology. Field spacing was 0.5 mT, and 191 FORCs were measured, with interaction field ($H_u$) between −15 and 15 mT and coercivity ($H_c$) from 0 to 60 mT. The maximum applied field was 1.4 T. The averaging time for each data point was 200 ms. The FORCinel software[56] was used for data processing and the VARIFORC algorithm[57] was used for smoothing ($S_{c0} = 4$, $S_{b0} = 3$, $S_{c1} = S_{b1} = 7$).

Previous rock magnetic data[12] are consistent with a fine-grained source of remanent magnetization, either biogenic or detrital single-domain magnetite. The AMS in a rock can be geometrically depicted as a triaxial ellipsoid minimum ($k_{min}$),

intermediate ($k_{int}$), and maximum ($k_{max}$) principal axes. Subsequent deformation associated with compaction after deposition modifies an original rock fabric. Previous AMS data at Apiro and Furlo reveal a minor clustering of $k_{min}$ axes and girdle of $k_{max}$ and $k_{int}$ axes, implying a mildly oblate compaction fabric in the plane of bedding[21]. Compared with previous AMS results for the Scaglia Rossa limestone[21], our new data from these same sections (Apiro and Furlo) confirm a minor oblate flattening in the plane of bedding generally for all samples. However, no correlation between AMS and inclination is found when the data are cross-plotted (Supplementary Fig. 6), suggesting that differential inclination shallowing cannot account for the systematic stratigraphic variation in inclination (Fig. 2).

Rock magnetic experiments highlight the presence of a non-interacting mid-coercivity phase interpreted to be magnetite, mixed with a higher coercivity phase in differing proportions interpreted as hematite (Supplementary Figs. 3 and 4). Six samples through the Apiro section (across the proposed TPW event) are very similar, dominantly magnetite with a small amount of hematite, while the one sample studied from Furlo has a higher proportion of hematite. Overall, our results agree with previous rock magnetic data[12] suggesting magnetite mixed with varying amounts of hematite. The Lowrie–Fuller test suggests that these phases are single domain or pseudo single domain, although results may be skewed by hematite for the sample from Furlo (Supplementary Fig. 3). ARM acquisition highlights the samples contain non-interacting particles close to the magnetotactic bacteria endmember suggesting the possibility of magnetofossils (Supplementary Fig. 3). The Fuller test suggests that the magnetization carried by the magnetite is detrital in origin, highlighting that any hematization did not also precipitate magnetite (Supplementary Fig. 3).

KappaBridge thermal susceptibility experiments on a sample from Apiro (Supplementary Fig. 4) show a minor drop in coercivity at 580 °C, the Curie temperature of magnetite, and an irreversible cooling curve with strong increase in susceptibility and a slight shift in Curie temperature. No Verwey transition is seen in the sample, but even a few weight percent titanium can cause the transition to be shifted to lower temperatures beyond the range of the Kappabridge[58], although the substitution of titanium would be surprising in biogenic magnetite.

The FORC distributions are dominated by the "central ridge" (Fig. 1c), which indicates that magnetofossils are the main constituent of the magnetic mineral assemblage. Biogenic magnetite, then, may be the answer to age-old mystery of what accounts for the remarkable stability of Scaglia Rossa remanent magnetization.

**Biostratigraphy**. Aliquots of palaeomagnetic samples from both Apiro and Furlo sections were prepared for biostratigraphic foraminifera identification. First occurrences and last occurrences were identified following a new age-calibrated biostratigraphic zonation[59]. Planktonic foraminifera were studied in washed residues. Preparation included gentle crushing, cold acetolysis with acetic acid (80%) following the method of Lirer[60], sieving through a 63 μm mesh, 1–2 h cleaning in an ultrasonic cleaner, and drying at 60 °C. The cold acetolysis method enables extraction of generally well-identifiable foraminifera even from indurated limestone. This offers the possibility of accurate taxonomic determination and detailed analysis of foraminiferal assemblages.

**Age models**. Age modeling of the sampled sections using dated magnetic reversal boundaries provides both time scales for each section and new biostratigraphic ages for global chronostratigraphic correlation (Supplementary Fig. 6). The Apiro and Furlo sections both span the C34n-C33r-C33n interval, where C34n-C33r is preserved at Furlo, C33r-C33n is present at both sections, and the top of C33n is identified at Apiro. At Apiro, we identify the C33r-C33n reversal within less than a half of a meter (Supplementary Fig. 7), but a substantial covered interval (from 50 to 83 m) masks the C34n-C33r reversal (Supplementary Figs. 2 and 7). At Furlo, the C34n-C33r reversal is documented within less than a meter (Supplementary Fig. 7). Following convention, ages for stratigraphic heights occurring between two dated levels are determined by linear interpolation, whereas ages of strata above and below dated levels are determined by linear extrapolation. The assumption of constant sediment accumulation rates is applied for each bracketed time segment.

Updated ages are used for the top and bottom of C33r, requiring recalculation of foraminiferal zonal boundary ages within chron 33. A refined age of 80.32 Ma for the C33r-C33n reversal boundary from U/Pb ages on zircon in bentonites in the Pierre Shale is used[61]. The base of C33r, the end of the Cretaceous long normal superchron, is radiometrically unconstrained[62], but cyclostratigraphy provides an astronomical calibration[37]. Ages of biostratigraphic boundaries are updated in Petrizzo et al.[59]; ages of those zonal boundaries that are calibrated by our work are provided in Supplementary Fig. 6.

Age models for each section are shown in Supplementary Fig. 6. A rather uniform sediment accumulation rate of ~10 m Myr⁻¹ is thought to characterize much of the pelagic Umbria–Marche basin[63]. Dating of our sections generally supports this classic interpretation. Compared with Gubbio, or any other section for that matter, Apiro is stratigraphically expanded. Increased sediment accumulation rates may indicate that the eastern thrust in which Apiro is exposed (Supplementary Fig. 1) represented an intrabasinal depocenter that sourced turbidite events recorded elsewhere (e.g., Furlo[39,64]).

**Local tectonic restoration of northern Umbria.** The Umbrian path of palaeo-poles has traditionally been regarded to have an "African" character: once relative rotation during Neogene time is restored, the two APW paths are nearly identical[65,66]. Northern Umbria is thought to have rotated counterclockwise relative to southern Umbria (and stable Africa) since Mesozoic time. Umbrian tectonic deformation is attributable to oroclinal bending and/or local or regional tectonic rotation about a sub-vertical axis in the Apennines[67,68]. van der Voo[66] follows Vandenburg et al.[69] who propose a 25° clockwise restoration of Northern Umbria into an African reference frame. Independently, Channell et al.[70] proposed a slightly smaller restoration of 22° clockwise, and subsequently rounded that estimate to 20° (ref. [65]). Most recently, van Hinsbergen et al.[71] developed a kinematic model for Mediterranean tectonic evolution using Euler rotations guided by palaeomagnetic and other datasets. For 80 Ma, northern Umbria (their plate #3980) restores to Africa with these Euler parameters: 64.75°N, 9.89°E, 4.01°CW. Within the northern Umbrian plate, local rotations are additionally possible. By comparing our declination means to the predicted values from van Hinsbergen et al.[71] and the global APW path of Torsvik et al.[72], we find minimal offset when Apiro is restored an additional +5°CW about a local axis, and Furlo +12°CW (Fig. 2). Our results add to abundant palaeomagnetic evidence for the orocline hypothesis of a near-linear belt that has been secondarily bent[73]. After we restore the palaeopoles from the Apennines to Africa with these rotations, we then apply the South Africa plate reconstruction to the palaeomagnetic reference frame for 80 Ma[72] to test the TPW hypothesis (Fig. 3). Also in the comparison of the Apiro and Furlo records (Fig. 2), Furlo inclinations are uniformly lower than those of Apiro, implying more inclination shallowing at Furlo than Apiro, and consistent with the occurrence of stylolitic pseudo-bedding at the former but not the latter (Supplementary Fig. 2). A flattening factor of 0.85 was applied to Furlo inclinations, a value that optimally matches inclinations from the two sections where they overlap in time (Supplementary Data 1). This flattening factor is similar to elongation/inclination analysis[74] applied to the Furlo dataset (Supplementary Fig. 10).

## Data availability

The palaeomagnetic data from this study are available in the online content of this paper and from the Open Science Framework (https://osf.io/hqzdy/?view_only=2c5ab4f569d842b8a7678b8bfd1a27dd). Magnetometer measurements are available from the corresponding author upon reasonable request.

## Code availability

Rock magnetic data were analyzed using the RAPID Matlab scripts[17] available at https://sourceforge.net/projects/paleomag/.

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

## Acknowledgements

Alessandro Montanari of the Osservatorio Geologico di Coldigioco provided invaluable field support and local geological knowledge. David Bice helped identify the critical outcrop at Apiro Dam. Assistance in the field was provided by Alec Brenner, Emily Carter, Stefan Farsang, Harry Golash, Hima Gudipati, Jessica Heringer, Harrison Miller, Alessandro Montanari, Karen Paczkowski, and Brian Penserini. Laboratory measurements were conducted by Jacob Abrahams, Michael Grappone, Garima Gupta, Marco Cruz-Heredia, Valerie Pietrasz, Kirby Sikes, and Lewis Ward. R.N.M. and J.L.K. were supported by NSF Geophysics grant EAR-1114432. R.N.M. was also supported by the the National Natural Science Foundation of China (No. 41890833 and 41772192) and the Institute of Geology & Geophysics, CAS, grant no. IGGCAS-201905. This is a contribution to the International Geoscience Programme 648.

## Author contributions

R.N.M., C.J.T., D.A.D.E., and J.L.K. designed the study. R.N.M., C.J.T., S.P.S., R.C., and J.L.K. conducted the field work. R.C. conducted the biostratigraphy. S.P.S. conducted the rock magnetic experiments. T.Y. conducted the FORC experiment. R.N.M., C.J.T., D.A.D.E., and S.P.S. conducted the palaeomagnetic and palaeogeographic analysis. R.N.M. wrote the manuscript with input from all authors.

## Competing interests

The authors declare no competing interests.
