## [Peer Review File · Nature Communications]

Editorial Note: Parts of this Peer Review File have been redacted as indicated where no third party permissions could be obtained.

REVIEWER COMMENTS

Reviewer #1 (Remarks to the Author):

Review of "A Late Cretaceous true polar wander oscillation" by R.N. Mitchell et al.

To test hypotheses about true polar wander, really good paleomagnetic data with lots of redundancy are required to track the wandering of the ancient spin-axis location. Many prior studies, especially some claiming that true polar wander does not occur, fall below such a standard. That is one reason why it is so refreshing to see this study with its abundant and beautiful paleomagnetic data. Mitchell et al. present an extensive paleomagnetic study of Late Cretaceous limestones from the Umbria region of Italy. They have used an impressive set of laboratory tools from modern paleomagnetism and rock magnetism to extract the magnetic memory of these rocks and show that they have a magnetic record very near the time of deposition of these samples. Figure 1 illustrates nice clean results from thermal demagnetization. The record shown in Figure 2 shows average magnetic directions every million years from 74 to 87 Ma determined from more than 1000 paleomagnetic samples. Beyond the superior quality of their data, the sheer number of paleomagnetic samples are orders of magnitude greater than in the classic study of Lowrie and Alvarez. Their paleomagnetic results contain orders of magnitude more information than the recent study published in Nature Communications by Bono et al. The weight of the numerous high-quality data gives great force to the results they obtain and many of the conclusions that they draw.

The specific prediction being tested by Mitchell et al is that true polar wander (TPW) occurred in Late Cretaceous time as proposed by Gordon (1983) and by Sager and Koppers (2000). Gordon (1983) found that Pacific plate apparent polar wander (APW) from ~90 Ma to ~81 Ma was orthogonal to the motion of the Pacific plate relative to the hotspots. He inferred a rotation of Pacific hotspots, which he attributed to TPW, of 13 ± 4 degrees. In the hotspot reference frame the spin axis generally shifted from longitudes near 120W to longitudes near 150E. With better age dates and a larger data set of seamount poles, Sager and Koppers (2001) found a slightly larger shift (16 to 21 great circle degs) in a shorter amount of time contained within the 90 to 81 Ma interval proposed by Gordon (1983).

Although it is shown in a different reference frame (i.e., the African reference frame rather than the hotspot reference frame), which makes comparison a little tricky, the shift from 84 Ma to 74 Ma shown by Mitchell et al. in Figure 3 is broadly similar in magnitude and orientation to those found by Gordon and by Sager and Koppers. The time interval of the shift is compatible with that found by Gordon (1983) because chron 33r, the young end of the shift he proposed, was in an upper bound on the age of the young end of the shift. Figure 2 shows that the shift is probably over at 78 Ma and it is great that Mitchell et al have five 1-Ma average results that demonstrate that the shift is indeed over at about 78 Ma, establishing a long-lived baseline for the spin axis position following the shift.

The title of the paper mentions a "TPW Oscillation" and so far in this review I have discussed only the younger portion of the proposed oscillation. From Figures 2 and 3, it can be seen that the earlier shift appears to have occurred from 87 Ma to 84 Ma, defined by just three 1-Ma averages and most strongly defined by the oldest (87 Ma) 1-Ma average. Neither Gordon (1983) nor Sager and Koppers (2001) found any evidence for such an earlier shift, but their data were not well situated to capture it in that time interval. The earlier (87-84 Ma) shift of Mitchell et al., while defined by good data, lacks the redundancy that captures the later (84-74 Ma) shift and also lacks a well-documented baseline of where the pole was before the shift. While the earlier shift is an important discovery, it is important for future work to try to delineate it better and to document the pre-shift baseline, perhaps from rock records in some other parts of the world. I think that the manuscript ought to give more emphasis to the fact that the younger part of the oscillation is far better documented in this new work than is the

older part of the oscillation.

I have a couple quibbles with the manuscript.

First, in paragraph 9 of page 8, the authors suggest that mantle viscosity may have been lower in the Cretaceous than at present. I do not think this is justified by their observed rate of TPW being ~25% higher than the supposed "speed limit" on the rate of TPW. Given the uncertainties in the parameters of mantle properties from which it is determined, the upper uncertainty in the "speed limit" is probably about 100%.

Second, in paragraph 10 of page 8, the authors compare their oscillation with large amplitude TPW "predicted" from so-called theoretical considerations from Steinberger et al. (2017). They say their results are "strikingly consistent" with the "theoretical predictions". My quick take on Steinberger et al. (2017) is that they "predict" a single large-amplitude (~90 great circle degree) shift, not a 12 great-circle-degree oscillation. So the prediction and observation do not seem consistent at all, much less "strikingly consistent". If I'm wrong about this, please clarify in the manuscript.

Despite a couple quibbles with the interpretation, I think the results from this hard-earned wonderful data set deserve publication in a widely read journal such as Science, Nature, or Nature Communication. With orders of magnitude more of much higher quality data, Mitchell et al. are able to handily reject the proposal of Cottrell and Tarduno (2000) that the Scaglia Rossa results preclude TPW. The new results provide considerable support to the proposal of Gordon (1983) and Sager and Koppers (2001) that a TPW shift >10 great circle degrees occurred in Late Cretaceous time. Moreover Mitchell et al. have a totally novel discovery (meriting further research) of an oscillation in true polar wander in Late Cretaceous time. It is path breaking work.

Below, I have a few minor suggestions and comments:

"Stylolitic" is misspelled throughout the manuscript.

Line 81: "of magnetite", not "a magnetite".

Line 134: Not obvious to me that "prolate" is right. If it is right, a little more explanation is needed. The non-hydrostatic moment of inertia is tri-axial. The hydrostatic moment of inertia is oblate. Please correct or clarify.

Line 163: The caption says the poles are from Apiro, but aren't they from both Apiro and Furlo?

2nd page of "Methods", 6 lines from the bottom, "entirety" is misspelled. It's probably even better to rewrite this: "Apiro contains all of the Scaglia Rossa Formation, except a 30-m-thick ...".

3rd page of "Methods", 6th line of "Paleomagnetism" section. "Remnant" is written where "Remanent" is meant.

8th page of "Methods", line 5: "Compared with" is called for rather than "Compared to".

9th page of "Methods", line 1: I think that you want "consistently", not "consistency".

Reviewer #2 (Remarks to the Author):

This manuscript addresses an interesting problem concerning small-amplitude changes in the position of the spin axis relative to the mantle (aka True Polar Wander, TPW). In this study, the authors relate a detailed paleomagnetic study of Late Cretaceous Italian limestones across several well-dated

magnetic reversals. They find that the inclinations and declinations shift with time, implying that either Italy shifted or spin axis shifted (as tracked by the paleomagnetic poles). Given the short time interval and the lack of appropriate European tectonic events, they conclude that the shift is likely TPW. The short time period is unusual in that most studies of TPW have limited resolution, so defining such a quick TPW wobble is difficult. It is possible here because the study is focused on a sedimentary section with a relatively high sedimentation rate.

There are two especially good reasons that make this study significant. First, it is rare to define a TPW wobble like this one. They have been hypothesized by theoretical studies, but I don't know of another that has been defined by data. Much larger inertial interchange TPW events have been documented but there the shift is quite large (90 degrees). The second reason is that this TPW event is an important data set in a debated topic - i.e., whether TPW has occurred in the past ~130 Myr. There are some influential authors who have stated that it has not.

The authors have done a commendable job at characterizing the magnetizations of the samples and checking to be sure that the observed changes in inclination and declination is not caused by some artifact. They examined inclination shallowing, which is a common problem with sediments, but they make a convincing argument that it would not cause the observed changes. They have examined the magnetic carrier grains with magnetic properties to argue that the magnetic remanence is recorded by small, non-interacting titanomagnetite grains, which is the most reliable type of recorder. All-in-all, the authors give sufficient background on the paleomagnetism to convince me that the romance is reliable.

My only quibble with the manuscript is that I would like to see the authors provide better context. This wobble is "hiding in plain sight" because this time interval has been examined by other paleomagnetic studies. Why did they miss it? I think that one answer is that any study that combines paleomagnetic poles from many locations will not have the resolution to see this wobble. Typically, those studies average over 10-20 Myr. I would particularly like to see the authors address Tarduno's paper. His Science comment on the Sager and Koppers paper is cited, but the authors never explain why Tarduno, also using data from Italian limestones, concluded that there was no shift. Furthermore, Tarduno and Smirnov (EPSL, 2001) claimed that there was no TPW during the past 130 Ma. There are some other papers that have come to the same conclusion. I would like to see this addressed with a short discussion.

The manuscript is generally well written. I flag a few rough spots in the typescript. The figures are well-done and informative.

The manuscript is in reasonably good shape and should be published after minor revisions.

Report Mitchell et al

The authors have obviously analysed a great number of samples and all aspects of the paleomagnetism seems fine with me, but the overall interpretation in terms of TPW I find problematic as discussed below.

Abstract General

Quite cleverly done to sell the paper by first stating that estimating TPW on Earth is much more complicated than other planets and moons (really?) due to plate tectonics and then introducing a great debate about TPW at around 84 Ma (really? Not much mentioned in the last decade). And then they confirm the earlier postulated TPW from the Pacific and then that this also fits forward subduction models (based on reference frame that don't see TPW at this time?).

Line 20: "...sections in Italy that provides evidence for a $\sim 12^\circ$ TPW oscillation from 86-78 Ma". Not real oscillation from **86 Ma**....it is the single point **87 Ma** that makes an **oscillation**?

Line 31: "...remagnetization and **tectonic structure**". The sampled region definitely has some structural/reconstruction challenges in order to compare the data with the rest of the world.

Line 33: "...hotly **debated**". NOT really.

Lines 90-92: "...Palaeomagnetic data from Italy must be locally restored relative to the larger African plate, which also must be tectonically reconstructed in order to test the TPW hypothesis (Methods)". That is not trivial and restoring to Adria and assume coherence with Africa is wrong (see van Hinsbergen et al. 2019).

Line 93: "...our new data show a $\sim 12^\circ$ oscillation ($\sim 24^\circ$ total) between 86-79 Ma (~ 7 million years)..". Again, does not implied oscillation depends only on the 87 Ma data-point?

Line 129-130: "amplitude of the $\sim 12^\circ \pm 3^\circ$ excursion of Italy palaeopoles (Fig. 3) furthermore overlaps within uncertainty the $16^\circ \pm 3^\circ$ dispersion of Pacific Plate palaeopoles" ...this is merely a statement and requires a diagram to show and what fits one use to relate Africa to the Pacific?

Lines 176-178: " Late Cretaceous TPW is nonetheless only documented for Italy, the Pacific Plate, and, with less confidence, a few other regions. The relative paucity of sampling is not surprising, however, provided the rapidity of the proposed event". This TPW event is actually only seen in Italy and Pacific and mean poles between 90 and 70 Ma, globally, does not fit either (see Fig. 3 redone a below)!

Figure 1

Yes latitudes and longitudes on the map but please list sampling co-ordinates somewhere (unless I missed it?)

Figure 2

Oscillating TPW very much relies on the oldest Apiro data; without it, the so-called "tectonic trend" would be inclinations just showing a steeping trend from 85 to 79 Ma and declinations showing a systematic clockwise rotation in roughly the same interval? Also plotted as poles, Apiro 87 Ma poles stand out as 86 and 84 Ma poles overlap in Figure 3b.

Figure 3 180E and OE should be switched ??

(a) Not sure what the authors plot here since it is not the original poles listed in Torsvik et al. (2008) and do not fit the poles listed in Extended Data Table 1. Since I had the newer Torsvik et al (2012) compilation handy in digital format, I plotted them below from 140 to 10 Ma

(b) States it is poles from **Apiro** restored to African reference frame and not based on **Furlo** data? Guess the authors first rotated the data to Adria but Adria (and Apulia) have definitely moved with respect to both Africa and Europe since the Cretaceous (see van Hinsbergen et al. 2019). I show the effect of that in the figure below BUT interestingly none of their poles come close to expected global means (90-70 Ma) --- so data seems anomalous with respect to the rest of the world?

References:

Torsvik et al. 2012. *Phanerozoic polar wander, paleogeography and dynamics. Earth Science Reviews* 114, 325-368.

van Hinsbergen et al. 2019. *Kinematic reconstruction and tectonic evolution of the Mediterranean region since the Triassic. Gondwana Research* <https://doi.org/10.1016/j.gr.2019.07.009>.

Extended Data Table 1

Poles are listed as a mixture of north (75-87 Ma) and south-poles (90-260 Ma) but the latter poles are not correctly plotted in Figure 3a (see above)

Responses in blue.

Revisions in red.

Line numbers cited refer to version of revised manuscript *without* tracked changes.

REVIEWER COMMENTS

Reviewer #1 (Remarks to the Author):

Review of “A Late Cretaceous true polar wander oscillation” by R.N. Mitchell et al.

To test hypotheses about true polar wander, really good paleomagnetic data with lots of redundancy are required to track the wandering of the ancient spin-axis location. Many prior studies, especially some claiming that true polar wander does not occur, fall below such a standard. That is one reason why it is so refreshing to see this study with its abundant and beautiful paleomagnetic data. Mitchell et al. present an extensive paleomagnetic study of Late Cretaceous limestones from the Umbria region of Italy. They have used an impressive set of laboratory tools from modern paleomagnetism and rock magnetism to extract the magnetic memory of these rocks and show that they have a magnetic record very near the time of deposition of these samples. Figure 1 illustrates nice clean results from thermal demagnetization. The record shown in Figure 2 shows average magnetic directions every million years from 74 to 87 Ma determined from more than 1000 paleomagnetic samples. Beyond the superior quality of their data, the sheer number of paleomagnetic samples are orders of magnitude greater than in the classic study of Lowrie and Alvarez. Their paleomagnetic results contain orders of magnitude more information than the recent study published in Nature Communications by Bono et al. The weight of the numerous high-quality data gives great force to the results they obtain and many of the conclusions that they draw.

The specific prediction being tested by Mitchell et al is that true polar wander (TPW) occurred in Late Cretaceous time as proposed by Gordon (1983) and by Sager and Koppers (2000). Gordon (1983) found that Pacific plate apparent polar wander (APW) from ~90 Ma to ~81 Ma was orthogonal to the motion of the Pacific plate relative to the hotspots. He inferred a rotation of Pacific hotspots, which he attributed to TPW, of 13 ± 4 degrees. In the hotspot reference frame the spin axis generally shifted from longitudes near 120W to longitudes near 150E. With better age dates and a larger data set of seamount poles, Sager and Koppers (2001) found a slightly larger shift (16 to 21 great circle degs) in a shorter amount of time contained within the 90 to 81 Ma interval proposed by Gordon (1983).

Although it is shown in a different reference frame (i.e., the African reference frame rather than the hotspot reference frame), which makes comparison a little tricky, the shift from 84 Ma to 74

Ma shown by Mitchell et al. in Figure 3 is broadly similar in magnitude and orientation to those found by Gordon and by Sager and Koppers. The time interval of the shift is compatible with that found by Gordon (1983) because chron 33r, the young end of the shift he proposed, was in an upper bound on the age of the young end of the shift. Figure 2 shows that the shift is probably over at 78 Ma and it is great that Mitchell et al have five 1-Ma average results that demonstrate that the shift is indeed over at about 78 Ma, establishing a long-lived baseline for the spin axis position following the shift.

We are pleased that the reviewer, having himself been involved in the ca. 84 Ma TPW debate, views our new dataset as a significant contribution to resolving and advancing it further.

The title of the paper mentions a “TPW Oscillation” and so far in this review I have discussed only the younger portion of the proposed oscillation. From Figures 2 and 3, it can be seen that the earlier shift appears to have occurred from 87 Ma to 84 Ma, defined by just three 1-Ma averages and most strongly defined by the oldest (87 Ma) 1-Ma average. Neither Gordon (1983) nor Sager and Koppers (2001) found any evidence for such an earlier shift, but their data were not well situated to capture it in that time interval. The earlier (87-84 Ma) shift of Mitchell et al., while defined by good data, lacks the redundancy that captures the later (84-74 Ma) shift and also lacks a well-documented baseline of where the pole was before the shift. While the earlier shift is an important discovery, it is important for future work to try to delineate it better and to document the pre-shift baseline, perhaps from rock records in some other parts of the world. I think that the manuscript ought to give more emphasis to the fact that the younger part of the oscillation is far better documented in this new work than is the older part of the oscillation.

Response: We appreciate the reviewer’s informed perspective. Both Reviewers 1 and 3 note that the basis for the older portion of the oscillation is not as substantial as that of the well-documented and replicated younger portion. We thus admit that Reviewer 1’s suggestion of being more open about this difference is a wise suggestion and beneficial to future work in Italy or elsewhere in the world that wishes to test the proposed oscillation.

We nonetheless do note that the oldest 1 Myr average (admittedly from only one of the two sections) is still based on $n = 39$ samples spanning $\gg 10,000$ years, i.e., sufficiently adequate for averaging geomagnetic secular variation and calculating a robust palaeomagnetic pole. (Thus, contrary to the claim of Reviewer 3 that the older portion of the oscillation is based on “one data point”, it is the average of 39 high-quality data.) Furthermore, the oldest 1 Myr average yields both the expected declination and inclination (palaeolatitude) for Italy at this age, that is, before and after the TPW excursion. Contrary to the criticisms of Reviewers 1 and 3, then, we would argue that there is thus a firm basis for the pre-shift baseline as well as a post-shift baseline (i.e., a “stable pole position” mentioned by Reviewer 1).

Revision: While still arguing firmly for an oscillation (noted by both Reviewers 1 and 2 to be a highly novel contribution), we have also revised the manuscript to note that the older portion of the oscillation arguably requires further testing to be globally confirmed. The added text reads (lines 184-186):

“It should also be noted that the older portion of the TPW oscillation depicted here has only been documented in Italy and thus requires additional testing in particular.”

I have a couple quibbles with the manuscript.

First, in paragraph 9 of page 8, the authors suggest that mantle viscosity may have been lower in the Cretaceous than at present. I do not think this is justified by their observed rate of TPW being ~25% higher than the supposed “speed limit” on the rate of TPW. Given the uncertainties in the parameters of mantle properties from which it is determined, the upper uncertainty in the “speed limit” is probably about 100%.

Response: We understand the reviewer’s point here. The TPW “speed limit” is rate-limited, to first order, by the viscosity of the lower mantle, which has a considerable orders-of-magnitude uncertainty. Nonetheless, independent of our fast TPW rate reported here (and elsewhere from the Pacific Plate), there is petrological evidence for mantle temperatures in the Late Cretaceous being notably higher than those of today¹. But we generally agree with Reviewer 1 that this relatively ambiguous implication of our observed TPW rate should not be overstated.

Revision: We have revised this portion of the manuscript to place less emphasis on an interpretation that, although viable and with independent corroborating evidence, is subject to large uncertainties. The revised text below (lines 144-149) leads into the following paragraph much better now:

“Despite consistency between the Late Cretaceous petrological¹ and palaeomagnetic data, the large uncertainties in the parameters of the mantle properties from which the TPW speed limit is determined² renders the implication for a hotter mantle at this time as constrained by TPW rate ambiguous at best. Instead, older, and potentially faster TPW events³, should be used to constrain such a problem. The slightly above average speed of ca. 84 Ma TPW could rather be explained by a larger mantle convective forcing than normal.”

Second, in paragraph 10 of page 8, the authors compare their oscillation with large amplitude TPW “predicted” from so-called theoretical considerations from Steinberger et al. (2017). They say their results are “strikingly consistent” with the “theoretical predictions”. My quick take on

Steinberger et al. (2017) is that they “predict” a single large-amplitude (~90 great circle degree) shift, not a 12 great-circle-degree oscillation. So the prediction and observation do not seem consistent at all, much less “strikingly consistent”. If I’m wrong about this, please clarify in the manuscript.

Response: On one hand, we must contend that the age of our oscillation based on data and that of the modeled shift based on subduction history is indeed strikingly consistent. On the other hand, we agree that the amplitude of the two TPW events is starkly different and one is an oscillation and the other is a single shift. We thus generally agree with Reviewer 1 that the discussion of this paragraph could place a slightly different emphasis on the comparison between our data and previous TPW modeling.

Revision: The text has been appended to reflect this more nuanced comparison between our observations of TPW and modeled TPW based on subduction history. The entire paragraph has been restructured and revised accordingly (lines 151-170):

“Our observed TPW oscillation of ~12° amplitude at this age can be used to parameterize models to better assess Earth’s nonhydrostatic figure and the relative effects of sinking slabs and rising mantle plumes on Earth’s rotational stability. Using the well-constrained history of recently subducted slabs, the modelled effect on Earth’s nonhydrostatic geoid that controls the location of the spin axis (and thus TPW) predicts a large and rapid TPW event between 80-90 Ma⁴, precisely when our observations indicates TPW in fact occurred (Figs. 2 and 3). This general agreement thus begins to reconcile the apparent discrepancy between TPW observations and the model predictions at this age⁴. Nonetheless, despite the striking consistency in suggesting large-scale (> 10°) TPW between 80-90 Ma, there is a notable contrast in the amplitudes and styles of the observed and predicted TPW events. Our data suggest a ~12° TPW oscillation, whereas the model⁴ predicts a ~90° TPW shift. Since the model considers only TPW excitation from subduction⁴, we speculate that plumes rising through the mantle at this time may have counteracted the effect of sinking slabs and thereby reduced the amplitude of TPW excited by subduction alone as plumes have the opposite effect of slabs on Earth’s geoid kernel⁴. Plumes leaving the core-mantle boundary (e.g., the Réunion hotspot later producing the ca. 66 Ma Deccan Traps) or those reaching the surface (e.g., the Marion hotspot producing the 84-91 Ma Morondova large igneous province) at this age, both of deep mantle origins and occurring in the Indian Ocean of the southern hemisphere⁵, would have been well positioned to have potentially counteracted the effect of subducting slabs largely in the North Pacific⁴. Such complications aside, on both a theoretical and empirical basis, Late Cretaceous TPW represents the youngest known large-scale TPW event in Earth history³.”

Despite a couple quibbles with the interpretation, I think the results from this hard-earned wonderful data set deserve publication in a widely read journal such as Science, Nature, or

Nature Communication. With orders of magnitude more of much higher quality data, Mitchell et al. are able to handily reject the proposal of Cottrell and Tarduno (2000) that the Scaglia Rossa results preclude TPW. The new results provide considerable support to the proposal of Gordon (1983) and Sager and Koppers (2001) that a TPW shift >10 great circle degrees occurred in Late Cretaceous time. Moreover Mitchell et al. have a totally novel discovery (meriting further research) of an oscillation in true polar wander in Late Cretaceous time. It is path breaking work.

We are very pleased to see the reviewer's positive feedback and encouragement.

Below, I have a few minor suggestions and comments:

“Stylolitic” is misspelled throughout the manuscript.

Revision: Corrected throughout the manuscript.

Line 81: “of magnetite”, not “a magnetite”.

Revision: Corrected.

Line 134: Not obvious to me that “prolate” is right. If it is right, a little more explanation is needed. The non-hydrostatic moment of inertia is tri-axial. The hydrostatic moment of inertia is oblate. Please correct or clarify.

Response: We appreciate the reviewer picking up on this detail. The nonhydrostatic Earth can have either a triaxial or nearly prolate shape, depending on long-wavelength mantle convection^{3,6}.

Revision: The revised text, including appropriate references, now reads (lines 128-129):

“The TPW axis (I_{min}) is defined by the triaxial or nearly prolate shape of the nonhydrostatic Earth due to long-wavelength mantle convection^{3,6}...”

Line 163: The caption says the poles are from Apiro, but aren't they from both Apiro and Furlo?

Response: Both Reviewers 2 and 3 wondered whether the poles plotted in Figure 3b were from just Apiro or both sections. To first order, the records between Apiro and Furlo are consistent with each other as demonstrated in Figure 2. Nonetheless, due to appreciable uncertainties in the different local tectonic restorations and degrees of inclination shallowing in the two sections, we opted not to combine the data from the two sections when calculating palaeopoles. Instead, we

use predominantly poles from Apiro with the exception of the 86 and 85 Ma poles from Furlo that fill the stratigraphic gap at Apiro (Supplementary Fig. 2; Supplementary Table 1). The two poles used from Furlo notably plot along a path connecting the 87 and 84 Ma poles of Apiro, thus, they are consistent. This explanation has been appended to the Methods section on “Palaeomagnetism”.

Revision: To further clarify what is plotted in Figure 3, we have added the following sentence to the figure legend (lines 455-457):

“All Italian poles are from Apiro except 85 and 84 Ma from Furlo that fill the gap at Apiro (Supplementary Fig. 2).”

2nd page of “Methods”, 6 lines from the bottom, “entirety” is misspelled. It’s probably even better to rewrite this: “Apiro contains all of the Scaglia Rossa Formation, except a 30-m-thick ...”.

Revision: We have revised the sentence using the reviewer’s suggestion.

3rd page of “Methods”, 6th line of “Paleomagnetism” section. “Remnant” is written where “Remanent” is meant.

Revision: Corrected.

8th page of “Methods”, line 5: “Compared with” is called for rather than “Compared to”.

Revision: Corrected.

9th page of “Methods”, line 1: I think that you want “consistently”, not “consistency”.

Revision: Corrected.

Richard Gordon

We are very glad that the informed reviewer sees the value of our work and we greatly appreciate his constructive comments.

Reviewer #2 (Remarks to the Author):

This manuscript addresses an interesting problem concerning small-amplitude changes in the position of the spin axis relative to the mantle (aka True Polar Wander, TPW). In this study, the

authors relate a detailed paleomagnetic study of Late Cretaceous Italian limestones across several well-dated magnetic reversals. They find that the inclinations and declinations shift with time, implying that either Italy shifted or spin axis shifted (as tracked by the paleomagnetic poles). Given the short time interval and the lack of appropriate European tectonic events, they conclude that the shift is likely TPW. The short time period is unusual in that most studies of TPW have limited resolution, so defining such a quick TPW wobble is difficult. It is possible here because the study is focused on a sedimentary section with a relatively high sedimentation rate.

There are two especially good reasons that make this study significant. First, it is rare to define a TPW wobble like this one. They have been hypothesized by theoretical studies, but I don't know of another that has been defined by data. Much larger inertial interchange TPW events have been documented but there the shift is quite large (90 degrees). The second reason is that this TPW event is an important data set in a debated topic - i.e., whether TPW has occurred in the past ~130 Myr. There are some influential authors who have stated that it has not.

The authors have done a commendable job at characterizing the magnetizations of the samples and checking to be sure that the observed changes in inclination and declination is not caused by some artifact. They examined inclination shallowing, which is a common problem with sediments, but they make a convincing argument that it would not cause the observed changes. They have examined the magnetic carrier grains with magnetic properties to argue that the magnetic remanence is recorded by small, non-interacting titanomagnetite grains, which is the most reliable type of recorder. All-in-all, the authors give sufficient background on the paleomagnetism to convince me that the remanence is reliable.

We are pleased to see that the reviewer sees the value in our high-resolution palaeomagnetic dataset as a means of not only testing ca. 84 Ma TPW, but also asserting empirical evidence for a ~12° TPW oscillation, or “wobble”, at that time.

My only quibble with the manuscript is that I would like to see the authors provide better context. This wobble is "hiding in plain sight" because this time interval has been examined by other paleomagnetic studies. Why did they miss it? I think that one answer is that any study that combines paleomagnetic poles from many locations will not have the resolution to see this wobble. Typically, those studies average over 10-20 Myr. I would particularly like to see the authors address Tarduno's paper. His Science comment on the Sager and Koppers paper is cited, but the authors never explain why Tarduno, also using data from Italian limestones, concluded that there was no shift. Furthermore, Tarduno and Smirnov (EPSL, 2001) claimed that there was no TPW during the past 130 Ma. There are some other papers that have come to the same conclusion. I would like to see this addressed with a short discussion.

Response: The last sentence of the penultimate paragraph implies why this “wobble” may have been “hiding in plain sight” (lines 182-186): “Future testing of coeval stratigraphies around the world should be conducted at commensurate resolution on palaeomagnetic targets of comparable quality.” That is, we suspect that aliasing is the main reason the wobble has been missed. Even though, as the reviewer correctly states, that this time interval has been examined by other palaeomagnetic studies, their sampling resolutions were not adequate to identify the $\sim 12^\circ$ TPW oscillation. As noted by Reviewer 1, our study sampled an order of magnitude more samples than typically done over a 10 Myr interval: $\sim 1,000$ samples instead of ~ 100 as done in typical, high-quality palaeomagnetic studies.

The manuscript presently contains an explanation for “why Tarduno, also using data from Italian limestones, concluded that there was no shift”. We agree with the reviewer that this is important to make clear in our paper the presents a resampling these rocks and would refer them to lines 35-42:

“It has been asserted, however, that classic palaeomagnetic data from the correlative Gubbio and Moria sections of Scaglia Rossa limestone do not permit ca. 84 Ma TPW⁷. Irrespective of the quality of the original data, an analysis that calculates only 3 average inclinations from 90-75 Ma⁷ does not constitute a robust test of a comparatively rapid² process. But the reliability of those old (however seminal) data^{8,9} are questionable¹⁰. In particular, the Gubbio and Moria data pre-date least-squares analysis of palaeomagnetic data¹¹, the use of controlled-atmosphere thermal demagnetization, and the measurement by sensitive superconducting magnetometers.”

The reviewer’s final valid point we did not already address: the general claim of a lack of TPW in the past 130 Myr. Concerning Tarduno and Smirnov¹² specifically, despite that paper’s declarative title claiming stability of the spin axis over the past 130 Myr, their analysis actually only addresses testing a $\sim 18^\circ$ TPW shift hypothesized by Prevot et al.¹³ to occur at ca. 110 Ma. In their test of Prevot et al.’s hypothesis, Tarduno and Smirnov¹² exclusively use data from one continent that Prevot et al.¹³ exclude. It is beyond the scope of our manuscript to weigh in on which data selection approach is valid or to assess the validity of an older TPW hypothesis that we did not study.

Nonetheless, there are other prominent authors (Trond Torsvik and Bernhard Steinberger) that despite arguing for multiple TPW events between 300-100 Myr ago, have also not identified any such events in the past 100 Myr^{14,15}. Precisely as the reviewer suspects, it is likely an artifact of such studies by these authors employing both 10-Myr averaging of palaeomagnetic data and using data from multiple continents to generate global apparent polar wander paths. Indeed, TPW should be a global phenomenon, which is why such an approach is taken. But while this method may be adequate for identifying larger amplitude TPW shifts ($20-30^\circ$) occurring over longer amounts of time (≥ 10 Myr)^{14,15}, it would smooth out and/or obscure the signal of a

smaller amplitude TPW shift ($\sim 10^\circ$) over a shorter amount of time (< 10 Myr). By contrast, our study employs 1-Myr averaging for one continent. Although TPW for the past 100 Myr may be of smaller amplitude than earlier Mesozoic TPW shifts^{14,15}, our high-resolution palaeomagnetic study would suggest that one cannot rule out the possibility of rapid TPW shifts of small amplitude ($< 20^\circ$) in the past 100 Myr.

Revision: As requested by the reviewer, we have appended a more thorough discussion of why papers suggesting that TPW is limited over the past 100 Myr may have yielded false negatives. The main reasons are over-averaging and/or the individual datasets themselves alias the possibility of a small, but significant, TPW shift occurring in < 10 Myr. We try to explain the relation of our work to past studies as sufficiently and briefly as possible in this entirely new paragraph that now concludes our discussion (lines 188-202):

“Even though this time interval has been examined by other palaeomagnetic studies, their sampling resolutions were likely insufficient to identify the TPW oscillation and thus aliased it. Our study collected an order of magnitude more samples than is typically done in high-quality palaeomagnetic studies over a 10 Myr interval, i.e., $\sim 1,000$ samples instead of ~ 100 . Despite arguing for multiple TPW events between 300-100 Ma (Fig. 3a), global appraisals of TPW have not identified any TPW events in the past 100 Myr^{22,33}. However, such analyses employ both 10-Myr averaging of palaeomagnetic data and use data from multiple continents to generate global apparent polar wander paths. Indeed, TPW should be a global phenomenon, which is why such an approach is taken. Nonetheless, while this method may be adequate for identifying larger amplitude TPW shifts ($20\text{-}30^\circ$) occurring over longer amounts of time (≥ 10 Myr), it would smooth out and/or obscure the signal of a smaller amplitude TPW shift ($\sim 10^\circ$) over a shorter amount of time (≤ 10 Myr). By contrast, as allowed by our increased sampling resolution, our study employs 1-Myr averaging of data from one continent. Although TPW for the past 100 Myr may be of smaller amplitude than earlier TPW shifts²⁴, our high-resolution palaeomagnetic study would suggest that one cannot rule out the possibility of rapid TPW shifts of small amplitude ($< 20^\circ$) in the most recent past.”

The manuscript is generally well written. I flag a few rough spots in the typescript. The figures are well-done and informative.

PDF comments appended below.

Line 15: unclear. The satellites are probably not on Earth

Revision: The sentence has been rewritten to be more precise (lines 14-15):

“True polar wander (TPW), or planetary reorientation, is well documented for other planets and moons and for Earth at present day with satellites...”

Line 44: Wording could be clearer. How about contemporaneous?

Revision: We now more clearly refer to Apiro and Furlo as “*stratigraphically correlative*” sections.

Line 52: Why is that significant?

Revision: The revised text now explains the geological significance of these observations (lines 60-63):

“Furthermore, stylonitic “pseudo-bedding” that characterizes the bedding style of the Scaglia Rossa at Gubbio and elsewhere in the Umbria-Marche basin¹⁶ are absent at Apiro, and less pronounced at Furlo (Supplementary Fig. 2), suggesting the sedimentary rocks of our sections are less tectonically modified than others.”

Line 66: units on this scale?

Revision: Units have been added to the scale bar of Figure 1c.

Line 93: Yes, 12 degrees in inclination. The reader will be thinking paleolatitude, which is only about half as much. Perhaps insert the approximate paleolatitude change in parenthesis.

Revision: In fact, this angular distance was indeed estimated in pole space (as the reviewer suggests it should be), which we now make clear by adding reference to Figure 3 after this statement.

Line 98: Unclear.

Revision: The revised text includes further explanation (lines 99-102):

“Furthermore, our demagnetization methods are detailed enough to resolve and remove overprints (Fig. 1a, b; Supplementary Fig. 3) and directional transitions observed across magnetic reversals are smooth instead of sudden (Fig. 2), as would be the case if directional shifts were artefacts of unresolved normal polarity overprints.”

Line 129: I'm a little confused. The change in Italy inclination and declination appears to be ~12 deg from Fig. 2. But here you are describing poles, so there should be a factor of about 2 less.

Response: The change in Italy's location affects (or is expressed by changes in) *both* inclination and declination. If one only considers one of these components alone (as done in Figure 2), then one underestimates the total angular offset. This is why the angular distance of the oscillation is estimated in pole space (Fig. 3b; addressed above), which is a function of both inclination and declination.

Line 139: This sentence is a bit hard to follow

Revision: The complicated sentence has been revised such that the previously mentioned details of TPW dynamics have been deleted and can be found in the references provided. The revised text now reads (lines 133-135):

“Also, as observed elsewhere⁵ and in high-resolution detail here (Figs. 2 and 3), TPW is typically modeled as a back-and-forth “roundtrip” oscillation where the pole shifts away, but then snaps backs to the original pole position²³.”

Line 172: unclear. This second dependent clause references "both" but subjects are not clear.

Revision: The text has been revised to clarify the subjects (lines 175-179):

“Palaeopoles of this age from Europe and North America, continents that are both well-resolved palaeogeographically with restoration of Atlantic seafloor spreading, both circumscribe polar excursions of similar magnitude as well as similar coplanar orientation to Italy, preliminarily passing the global reproducibility test of the TPW hypothesis (Supplementary Fig. 9; ref. ⁹).”

Line 175: unclear

Revision: The lack of clarity was due to a typo; the word “of” has been replaced by “or”. The revised text now reads (lines 179-181):

“Unlike earlier Mesozoic TPW events that are substantiated by the palaeomagnetic records of multiple or even most continents^{22,29-32}, Late Cretaceous TPW is nonetheless only documented for Italy, the Pacific Plate, and, with less confidence, a few other regions.”

Methods, Page 2: are

Revision: Corrected.

Methods, Page 4: How many? Presumably this was a subset and not all.

Revision: Per the reviewer's suggestion, the revised text reads (lines 283-284):

“Non-destructive rock magnetic experiments were performed on 7 samples of the Scaglia Rossa limestone (6 samples from Apiro and 1 sample from Furlo) ...”

Methods, Page 6 (Rock magnetism): Lowrie-Fuller?

Response: As we have indicated, we do in fact mean the Fuller test, which is an empirically calibrated test (for magnetite) to understand the origin of magnetization in a sample^{23,24} and is explained in its figure legend.

Methods, Page 6 (Biostratigraphy): LOs

Revision: Corrected.

Methods, Page 11: change in font

Revision: Font fixed.

Methods, Page 20: typo

Revision: Corrected.

The manuscript is in reasonably good shape and should be published after minor revisions.

We appreciate the reviewer's detailed editorial comments and constructive questions on how to relate our new data to the context of prior work on TPW in the past 100 Myr.

Reviewer #3 (Remarks to the Author):

See enclosed PDF file

PDF comments appended below.

Report Mitchell et al

The authors have obviously analysed a great number of samples and all aspects of the paleomagnetism seems fine with me, but the overall interpretation in terms of TPW I find problematic as discussed below.

We are pleased that the reviewer appreciates the significant amount of work we have done and are very pleased to hear that the data seem solid in their opinion. The reviewer is of course entitled to a difference of opinion regarding our TPW interpretation; nonetheless, we have done our best to address those aspects the reviewer views as problematic in hopes of better justifying our argument.

Abstract General

Quite cleverly done to sell the paper by first stating that estimating TPW on Earth is much more complicated than other planets and moons (really?) due to plate tectonics and then introducing a great debate about TPW at around 84 Ma (really? Not much mentioned in the last decade). And then they confirm the earlier postulated TPW from the Pacific and then that this also fits forward subduction models (based on reference frame that don't see TPW at this time?).

Response: Estimating TPW on any planet or moon is admittedly not without controversy. Nonetheless, both the crustal reorientations and “membrane” stresses often interpreted as evidence of TPW on other planets and moons, when observed on Earth, must compete with tectonic interpretations and be deconvolved from plate tectonic crustal displacements and stresses.

We agree that the heated debate over the possibility of ca. 84 Ma TPW occurred over a decade ago. However, we would assert first that just because the debate occurred sometime ago, does not change the fact that the issue is important and unresolved and, second, that the lack of debate since likely reflects the severe degree of the prior stalemate. Furthermore, we would point out that Reviewer 1 acknowledges that the long-standing issue has already spanned several decades and Reviewer 2 agrees that the debate was intense and our presenting a solution to it represents a significant contribution.

Indeed, as the reviewer suspects, the reference frame that Steinberger et al.⁴ use in their forward modeling of TPW using past subduction histories does not identify TPW at this time, even though the subduction history in their forward modeling predicts there should have been. Thus, our observation of TPW at the same age as predicted by the forward subduction model begins to reconcile such theory and data.

Line 20: “...sections in Italy that provides evidence for a ~12° TPW oscillation from 86-78 Ma”.
Not real oscillation from 86 Ma....it is the single point 87 Ma that makes an oscillation?

Response: The oldest 1 Myr average (admittedly from only one of the two sections) is still nonetheless based on $n = 39$ samples spanning $\gg 10,000$ years, i.e., sufficiently adequate for averaging geomagnetic secular variation and calculating a robust palaeomagnetic pole. Thus, the older portion of the oscillation is not based on “one data point”, it is based on the average of 39 high-quality data.

Line 31: “...remagnetization and tectonic structure”. The sampled region definitely has some structural/reconstruction challenges in order to compare the data with the rest of the world.

Response: We of course agree with the point the reviewer is making (addressed in more detail below) as we include an entire section in the Methods discussing the structural/tectonic reconstruction of Italy. Nonetheless, in the context of the sentence in question, the point is that uncertainties related to such reconstructions are obviated by testing for TPW in a single stratigraphic section that itself is devoid of structural complications.

Line 33: “...hotly debated”. NOT really.

Response: Addressed above. As previously noted, both Reviewers 1 and 2 share a different opinion.

Lines 90-92: “...Palaeomagnetic data from Italy must be locally restored relative to the larger African plate, which also must be tectonically reconstructed in order to test the TPW hypothesis (Methods)”. That is not trivial and restoring to Adria and assume coherence with Africa is wrong (see van Hinsbergen et al. 2019).

Response: As indicated at the end of the sentence the reviewer quotes, our Methods include a detailed description of how we restore our Italian data from north Umbria to the African plate. We appreciate the reviewer pointing out the recent paper on this topic by van Hinsbergen et al. (2020)²⁵ that cites an earlier, but also recent, paper (van Hinsbergen et al. [2014]²⁶) concerning the question of whether Adria rotated relative to Africa. Citing the 2014 paper²⁶, the 2020 paper²⁵ calls for a 10° counter-clockwise rotation of Adria relative to Africa (implying that at least a clockwise rotation of this magnitude is required to restore our Italian palaeomagnetic data to Africa). As detailed in our Methods section entitled, “Local tectonic restoration of northern Umbria”, the clockwise rotations that we apply of 12° for Furlo and 5° for Apiro from north Umbria are quite consistent with Adria needing ~10° rotation to Africa²⁶.

Revision: We have appended multiple references to van Hinsbergen et al.²⁵ and discuss the constraints in the relevant Methods section. The revised text now reads (lines 376-399):

“Local tectonic restoration of northern Umbria. The Umbrian path of palaeopoles has traditionally been regarded to have an “African” character: once relative rotation during Neogene time is restored, the two apparent polar wander (APW) paths are nearly identical^{27,28}. Northern Umbria is thought to have rotated counterclockwise relative to southern Umbria (and stable Africa) since Mesozoic time. Umbrian tectonic deformation is attributable to oroclinal bending and/or local or regional tectonic rotation about a sub-vertical axis in the Apennines^{29,30}. Van der Voo²⁸ follows Vandenburg et al.³¹ who propose a 25° clockwise restoration of Northern Umbria into an African reference frame. Independently Channell et al.³² proposed a slightly smaller restoration of 22° clockwise, and has subsequently rounded that estimate to 20° (ref. ²⁷). Most recently, van Hinsbergen et al.²⁵ developed a kinematic model for Mediterranean tectonic evolution using Euler rotations guided by palaeomagnetic and other datasets. For 80 Ma, northern Umbria (their plate #3980) restores to Africa with these Euler parameters: 64.75°N, 9.89°E, 4.01°CW. Within the northern Umbrian plate, local rotations are additionally possible. By comparing our declination means to the predicted values from van Hinsbergen et al.²⁵ and the global APW path of Torsvik et al.³³, we find minimal offset when Apiro is restored an additional +5°CW about a local axis, and Furlo +12°CW (Fig. 2). Our results add to abundant palaeomagnetic evidence for the orocline hypothesis of a near-linear belt that has been secondarily bent³⁴. After we restore the palaeopoles from the Apennines to Africa with these rotations, we then apply the South Africa plate reconstruction to the palaeomagnetic reference frame for 80 Ma³³ for to test the TPW hypothesis (Fig. 3).”

Figures 2 and 3, along with their captions, have also been revised accordingly and are shown below in response to the most relevant reviewer comment.

Line 93: “..our new data show a ~12° oscillation (~24° total) between 86-79 Ma (~7 million years)..” Again, does not implied oscillation depends only on the 87 Ma data-point?

Response: Addressed above.

Line 129-130: “amplitude of the $\sim 12^\circ \pm 3^\circ$ excursion of Italy palaeopoles (Fig. 3) furthermore overlaps within uncertainty the $16^\circ \pm 3^\circ$ dispersion of Pacific Plate palaeopoles” ...this is merely a statement and requires a diagram to show and what fits one use to relate Africa to the Pacific?

Response: The statement concerns a comparison between the observed amplitudes (which are within uncertainty of each other) and does not require a diagram, but only the numbers as provided. The diagram the reviewer is requesting would be, we assume, to compare not only the amplitudes of the polar displacement, but also their orientations (i.e., longitude of the polar shift). This is already discussed in the text (lines 173-179):

“Although the Pacific Plate exhibits a polar shift of similar age and magnitude to Italy^{9,10}, uncertainty of the palaeogeographic reconstruction of the Pacific Plate precludes testing the consistency of the palaeolongitude of the TPW axis. Palaeopoles of this age from Europe and North America, continents that are both well-resolved palaeogeographically with restoration of Atlantic seafloor spreading, both circumscribe polar excursions of similar magnitude as well as similar coplanar orientation to Italy, preliminarily passing the global reproducibility test of the TPW hypothesis (Supplementary Fig. 9; ref. ⁹).”

Lines 176-178: “Late Cretaceous TPW is nonetheless only documented for Italy, the Pacific Plate, and, with less confidence, a few other regions. The relative paucity of sampling is not surprising, however, provided the rapidity of the proposed event”. This TPW event is actually only seen in Italy and Pacific and mean poles between 90 and 70 Ma, globally, does not fit either (see Fig. 3 redone a below)!

Response: As discussed in the cited text immediately above and shown in Supplementary Figure 9, evidence for ca. 84 Ma TPW is not limited to Italy and the Pacific plate. Also, as discussed below in response to the specific comment, the reviewer’s assertion that our data do not fit with global palaeopoles between 90 and 70 Ma is an artifact of the reviewer misunderstanding that our figure plots the poles in reconstructed coordinates at 80 Ma whereas they plotted the global poles in present-day coordinates. The potential for misunderstanding by other readers has been addressed (see responses to related specific reviewer comment below).

Both Reviewers 2 and 3 made the valid criticism that the ca. 84 Ma TPW event does not seem to be supported by previous assessments of the global paleomagnetic database at this age.

Revisions: Since other readers may have a similar question—how to reconcile this TPW event with previous global assessments that did not detect it—we have written an entirely new paragraph to the end of our discussion. This additional discussion further allows us to highlight why our high-resolution study is unprecedented in palaeomagnetism (alluded to by Reviewer 1). This new paragraph goes as follows (lines 188-202):

“Even though this time interval has been examined by other palaeomagnetic studies, their sampling resolutions were likely insufficient to identify the TPW oscillation and thus aliased it. Our study collected an order of magnitude more samples than is typically done in high-quality palaeomagnetic studies over a 10 Myr interval, i.e., ~1,000 samples instead of ~100. Despite arguing for multiple TPW events between 300-100 Ma (Fig. 3), global appraisals of TPW have not identified any TPW events in the past 100 Myr^{22,33}. However, such analyses employ both 10-Myr averaging of palaeomagnetic data and use data from multiple continents to generate global

apparent polar wander paths. Indeed, TPW should be a global phenomenon, which is why such an approach is taken. Nonetheless, while this method may be adequate for identifying larger amplitude TPW shifts (20-30°) occurring over longer amounts of time (≥ 10 Myr), it would smooth out and/or obscure the signal of a smaller amplitude TPW shift ($\sim 10^\circ$) over a shorter amount of time (≤ 10 Myr). By contrast, as allowed by our increased sampling resolution, our study employs 1-Myr averaging of data from one continent. Although TPW for the past 100 Myr may be of smaller amplitude than earlier TPW shifts²⁴, our high-resolution palaeomagnetic study would suggest that one cannot rule out the possibility of rapid TPW shifts of small amplitude ($< 20^\circ$) in the most recent past.”

Figure 1

Yes latitudes and longitudes on the map but please list sampling co-ordinates somewhere (unless I missed it?)

Revision: GPS coordinates for both sections have been added to the Methods section entitled “Geologic setting and sampling” (lines 218-220).

Figure 2

Oscillating TPW very much relies on the oldest Apiro data; without it, the so-called “tectonic trend” would be inclinations just showing a steeping trend from 85 to 79 Ma and declinations showing a systematic clockwise rotation in roughly the same interval? Also plotted as poles, Apiro 87 Ma poles stand out as 86 and 84 Ma poles overlap in Figure 3b.

Response: The oldest 1 Myr average (87 Ma palaeopole from Apiro) yields both the expected declination and inclination (palaeolatitude) for Italy at this age, that is, before and after the TPW excursion (or if TPW had not occurred). There is thus a firm basis in anchoring the tectonic trend to this oldest pole that yields the expected, not an anomalous, pole position. The revised “tectonic trend” suggested by Reviewer 3 would require: (i) discarding the 87 Ma pole (an average based on 39 high-quality data), for which there is no apparent justification particularly as the direction is that expected of Italy if TPW had not occurred and (ii) interpreting the anomalous data from 85 to 79 Ma as due to plate tectonics as suggested would violate the plate tectonic speed limit that is empirically and theoretically constrained to be ~ 20 cm yr⁻¹ (refs.^{36,37}) as these data imply rates of motion ≥ 30 cm yr⁻¹.

Response: We have also tried to revise Figure 2 to better illustrate then deviation we interpret as TPW from the expected plate motion. The relevant portion of the revised caption of Figure 2 (below) now reads (lines 443-448):

“Directional predictions for a representative site (43.5°N, 12.9°E) from the global apparent polar wander path of Torsvik et al.³³ and the tectonic model of van Hinsbergen et al.²⁵ are shown in both North Umbrian and South African coordinates. Apiro declination data have been adjusted for local rotation to match those reference frames, and Furlo data have been corrected to match optimally the Apiro dataset with local rotation and inclination shallowing³⁸.”

Figure 3

180E and 0E should be switched ??

(a) Not sure what the authors plot here since it is not the original poles listed in Torsvik et al. (2008) and do not fit the poles listed in Extended Data Table 1. Since I had the newer Torsvik et al (2012) compilation handy in digital format, I plotted them below from 140 to 10 Ma

(b) States it is poles from Apiro restored to African reference frame and not based on Furlo data? Guess the authors first rotated the data to Adria but Adria (and Apulia) have definitely moved with respect to both Africa and Europe since the Cretaceous (see van Hinsbergen et al. 2019). I show the effect of that in the figure below BUT interestingly none of their poles come close to expected global means (90-70 Ma) --- so data seems anomalous with respect to the rest of the world?

Response: 180°E and 0°E should not be switched. In the figure they generated, Reviewer 3 plotted data from Torsvik et al. (2012)³³ as north poles, but mislabeled the lines of longitude; note in this version of Torsvik et al. (2012)³³ shown below as south poles, the lines of longitude are labeled the same as in Reviewer 3's figure (i.e., with 0°E at the top, which should only be the case when plotting south poles). Our Figure 3 plots north poles, and so 0°E is correctly at the bottom of the plot.

[redacted]

(see Figure 10A of Doubrovine, P. V., Steinberger, B., and Torsvik, T. H., 2012, Absolute plate motions in a reference frame defined by moving hot spots in the Pacific, Atlantic, and Indian oceans: *Journal of Geophysical Research*, v. 117, p. B09101.)

(a) The poles plotted in Figure 3a and listed in Supplementary Table 3 are indeed from Torsvik et al. (2008)³⁹ as stated and we double-checked that the values are correct as listed and plotted. We also converted the poles from Torsvik et al. (2008)³⁹ into north poles (see below).

(b) Both Reviewers 2 and 3 wondered whether the poles plotted in Figure 3b were just from Apiro or both sections. To first order, the records between Apiro and Furlo are consistent with each other as demonstrated in Figure 2. Nonetheless, due to appreciable uncertainties in the different local tectonic restorations and degrees of inclination shallowing in the two sections, we opted not to combine the data from the two sections when calculating palaeopoles. Instead, we use exclusively poles from Apiro with the exception of the 85 and 84 Ma poles from Furlo that fill the gap at Apiro (Supplementary Fig. 2; Supplementary Table 1). The two poles imported from Furlo notably plot along a path connecting the 86 and 83 Ma poles of Apiro, thus, they are consistent. This explanation has been appended to the Methods section on “Palaeomagnetism”.

Revision: To further clarify what is plotted, we have added the following sentence to the Figure 3 legend (lines 455-457):

“All Italian poles are from Apiro except 85 and 84 Ma from Furlo that fill the gap at Apiro (Supplementary Fig. 2)”

Also, as addressed above in detail, directional predictions for a representative site (43.5°N, 12.9°E) from the global apparent polar wander path of Torsvik et al.³³ and the tectonic model of van Hinsbergen et al.²⁵ are shown in the revised Figure 2 (above) in both North Umbrian and South African coordinates. Apiro declination data have been adjusted for local rotation to match those reference frames, and Furlo data have been corrected to match optimally the Apiro dataset with local rotation.

In their figure above, Reviewer 3 has plotted the 90-70 Ma global palaeopoles in present-day coordinates, whereas our Figure 3 plots all palaeopoles in reconstructed coordinates at 80 Ma. The apparent mismatch of our data with global palaeopoles is due to this confusion.

Revision: To make sure other readers do not have the same confusion, we have made our plotting the palaeopoles in a reconstructed reference frame (i.e., not present-day coordinates) clearer in the Figure 3 legend (lines 454-455):

“Italy and its new poles poles (yellow) reconstructed using Euler parameters (4.0°N, 32.6°W, 21.8°CW) in a global plate model with simplified Mediterranean motions guided by ref. ²⁵.”

This is also made clearer in our revised Methods section (lines 391-393):

“After we restore the palaeopoles from the Apennines to Africa with these rotations, we then apply the South Africa plate reconstruction to the palaeomagnetic reference frame for 80 Ma³³ to test the TPW hypothesis (Fig. 3).”

Figure 3 has also been revised slightly (below) based on this updated local restoration of the Italian data. We also now use the updated apparent polar wander path of Torsvik et al.³³ as suggested by the reviewer.

References:

Torsvik et al. 2012. Phanerozoic polar wander, paleogeography and dynamics. *Earth Science Reviews* 114, 325-368.

van Hinsbergen et al. 2019. Kinematic reconstruction and tectonic evolution of the Mediterranean region since the Triassic. *Gondwana Research*

<https://doi.org/10.1016/j.gr.2019.07.009>.

Extended Data Table 1

Poles are listed as a mixture of north (75-87 Ma) and south-poles (90-260 Ma) but the latter poles are not correctly plotted in Figure 3a (see above)

Revision: The poles listed in Supplementary Table 3 are now all north poles.

Response: The poles in Figure 3 were plotted correctly. Hopefully our further clarification in our revisions to the figure legend (addressed above) avoid other readers having similar confusion.

We thank the reviewer for their constructive criticisms which helped improve the manuscript.

References

- 1 Herzberg, C. & Gazel, E. Petrological evidence for secular cooling in mantle plumes. *Nature* **458**, 619-622 (2009).
- 2 Tsai, V. C. & Stevenson, D. J. Theoretical constraints on true polar wander. *J. Geophys. Res.-Solid Earth* **112**, B05415 (2007).
- 3 Mitchell, R. N. True polar wander and supercontinent cycles: Implications for lithospheric elasticity and the triaxial Earth. *American Journal of Science* **314**, 966-979 (2014).
- 4 Steinberger, B., Seidel, M.-L. & Torsvik, T. H. Limited true polar wander as evidence that Earth's nonhydrostatic shape is persistently triaxial. *Geophysical Research Letters* **44**, 2016GL071937 (2017).
- 5 Glisovic, P. & Forte, A. M. On the deep-mantle origin of the Deccan Traps. *Science* **355**, 613-616 (2017).
- 6 Creveling, J. R., Mitrovica, J. X., Chan, N. H., Latychev, K. & Matsuyama, I. Mechanisms for oscillatory true polar wander. *Nature* **491**, 244-248 (2012).
- 7 Cottrell, R. D. & Tarduno, J. A. Late Cretaceous true polar wander: Not so fast (comment) *Science* **288**, 2283a (2000).
- 8 Lowrie, W. & Alvarez, W. Late Cretaceous geomagnetic polarity sequence: detailed rock and palaeomagnetic studies of the Scaglia Rossa limestone at Gubbio, Italy. *The Geophysical journal of the Royal Astronomical Society* **51**, 561-581 (1977).
- 9 Alvarez, W. & Lowrie, W. Upper Cretaceous palaeomagnetic stratigraphy at Moria (Umbrian Apennines, Italy); verification of the Gubbio section. *Geophysical Journal of the Royal Astronomical Society* **55**, 1-17 (1978).
- 10 Sager, W. W. & Koppers, A. A. P. Late Cretaceous true polar wander: Not so fast (response). *Science* **288**, 2283a (2000).
- 11 Kirschvink, J. L. The least-squares line and plane and the analysis of palaeomagnetic data. *Geophysical Journal of the Royal Astronomical Society* **62**, 699-718 (1980).
- 12 Tarduno, J. A. & Smirnov, A. V. Stability of the Earth with respect to the spin axis for the last 130 million years. *Earth and Planetary Science Letters* **184**, 549-553 (2001).
- 13 Prevot, M., Mattern, E., Camps, P. & Daignieres, M. Evidence for a 20° tilting of the Earth's rotation axis 110 million years ago. *Earth and Planetary Science Letters* **179**, 517-528 (2000).
- 14 Steinberger, B. & Torsvik, T. H. Absolute plate motions and true polar wander in the absence of hotspot tracks. *Nature* **452**, 620-623 (2008).

- 15 Torsvik, T. H. *et al.* Deep mantle structures as a reference from for movements in and on the Earth. *Proceedings of the National Academy of Sciences* **111**, 8735-8740 (2014).
- 16 Alvarez, W., Colacicchi, R. & Montanari, A. Synsedimentary Slides and Bedding Formation in Apennine Pelagic Limestones. *Journal Of Sedimentary Petrology* **55**, 720-734 (1985).
- 17 Maloof, A. C. *et al.* Combined paleomagnetic, isotopic, and stratigraphic evidence for true polar wander from the Neoproterozoic Akademikerbreen Group, Svalbard, Norway. *Geological Society Of America Bulletin* **118**, 1099-1124 (2006).
- 18 Sager, W. W. & Koppers, A. A. P. Late cretaceous polar wander of the Pacific plate: Evidence of a rapid true polar wander event. *Science* **287**, 455-459 (2000).
- 19 Muttoni, G. & Kent, D. V. Jurassic monster polar shift confirmed by sequential paleopoles from Adria, promontory of Africa. *Journal of Geophysical Research: Solid Earth* **124**, 3288-3306 (2019).
- 20 Fu, R. R., Kent, D. V., Hemming, S. R., Gutierrez, P. & Creveling, J. R. Testing the occurrence of Late Jurassic true polar wander using the La Negra volcanics of northern Chile. *Earth and Planetary Science Letters* **529**, 115835 (2020).
- 21 Kent, D. V., Kjarsgaard, B. A., Gee, J. S., Muttoni, G. & Heaman, L. M. Tracking the Late Jurassic apparent (or true) polar shift in U-Pb-dated kimberlites from cratonic North America (Superior Province of Canada). *Geochemistry Geophysics Geosystems* **16**, 983-994 (2015).
- 22 Fu, R. R. & Kent, D. V. Anomalous Late Jurassic motion of the Pacific Plate with implications for true polar wander. *Earth and Planetary Science Letters* **490**, 20-30 (2018).
- 23 Fuller, M., Kidane, T. & Ali, J. AF demagnetization characteristics of NRM, compared with anhysteretic and saturation isothermal remanence: an aid in the interpretation of NRM. *Physics and Chemistry of the Earth* **27**, 1169-1177 (2002).
- 24 Fuller, M. *et al.* NRM: IRM(S) demagnetization plots; An aid to the interpretation of natural remanent magnetization. *Geophysical Research Letters* **15**, 518-521 (1988).
- 25 van Hinsbergen, D. J. J. *et al.* Orogenic architecture of the Mediterranean region and kinematic reconstruction of its tectonic evolution since the Triassic. *Gondwana Research* **81**, 79-229 (2020).
- 26 van Hinsbergen, D. J. J. *et al.* Did Adria rotate relative to Africa? *Solid Earth* **5**, 611-629 (2014).
- 27 Channell, J. E. T. Paleomagnetic data from Umbria (Italy): implications for the rotation of Adria and Mesozoic apparent polar wander paths. *Tectonophysics* **216**, 365-378 (1992).
- 28 van der Voo, R. *Paleomagnetism of the Atlantic, Tethys, and Iapetus Oceans*. (Cambridge University Press, 1993).
- 29 Muttoni, G., Argnani, A., Kent, D. V., Abrahamsen, N. & Cibin, U. Paleomagnetic evidence for Neogene tectonic rotations in the northern Apennines, Italy. *Earth And Planetary Science Letters* **154**, 25-40 (1998).

- 30 Muttoni, G. *et al.* Paleomagnetic evidence for a Neogene two-phase counterclockwise tectonic rotation in the Northern Apennines (Italy). *Tectonophysics* **326**, 241-253 (2000).
- 31 Vandenberg, J., Klootwijk, C. T. & Wonders, A. A. H. Late Mesozoic and Cenozoic movements of the Italian Peninsula: Further paleomagnetic data from the Umbrian sequence. *Geological Society of America Bulletin* **89**, 133-150 (1978).
- 32 Channell, J. E. T., Lowrie, W., Medizza, F. & Alvarez, W. Paleomagnetism And Tectonics In Umbria, Italy. *Earth And Planetary Science Letters* **39**, 199-210 (1978).
- 33 Torsvik, T. H. *et al.* Phanerozoic polar wander, palaeogeography and dynamics. *Earth-Science Reviews* **114**, 325-368 (2012).
- 34 Carey, S. W. The Orocline Concept in Geotectonics. *Proceedings of the Royal Society Tasmania* **89**, 255-288 (1955).
- 35 Gordon, R. G. Late Cretaceous apparent polar wander of the Pacific Plate; evidence for a rapid shift of the Pacific hotspots with respect to the spin axis. *Geophysical Research Letters* **10**, 709-712 (1983).
- 36 Conrad, C. & Hager, B. Mantle convection with strong subduction zones. *Geophysical Journal International* **144**, 271-288 (2001).
- 37 Bevis, M. *et al.* Geodetic observations of very rapid convergence and back-arc extension at the Tonga arc. *Nature* **374**, 249-251 (1995).
- 38 King, R. F. Remanent magnetism of artificially deposited sediments. *Geophysical Supplements to the Monthly Notices of the Royal Astronomical Society* **7.3**, 115-134 (1955).
- 39 Torsvik, T. H., Müller, R. D., Voo, R. V. d., Steinberger, B. & Gaina, C. Global plate motion frames: Toward a unified model. *Reviews Of Geophysics* **46**, doi:doi: 10.1029/2007RG000227 (2008).

REVIEWERS' COMMENTS

Reviewer #1 (Remarks to the Author):

I think the authors have done a solid job of responding to the reviews, in particular to my prior comments. I do have one remaining comment that I didn't bring up in my prior review, but should have, that relates to lines 192-193 and elsewhere. Although global appraisals have not identified TPW events in the past 100 Ma, my group has identified a couple of events mainly from Pacific plate data (but we think they are recorded in the global data, but with much lower precision and resolution). First, we identified a shift of about 8 degrees in Eocene time (Petronotis, Gordon, and Acton, 1994, full citation below), which we hypothesized was caused by TPW. We have lots of recent abstracts strongly supporting that early inference (and a manuscript in revision for Nature Communications).

More recently, Woodworth and Gordon (2018) proposed that about 3 degrees of TPW has occurred since 12 Ma and is probably continuing to occur.

These events are smaller in amplitude than the Cretaceous event investigated by the authors, but the context of the manuscript is incomplete without mentioning them I think.

A few other comments and suggestions:

Line 16: Geodesists use the term "contemporaneous" to mean "now" or the past few years or decades. Better to use the term "coeval" I think.

Line 24: please insert "non-hydrostatic" in front of "inertia"

Line 39: Please clarify that "questionable" applies to testing for true polar wander, but not to reliability for reversal studies. Or, if you mean that too, be clear about it.

References mentioned above:

```
@article{petronotis199457,  
title={A 57 Ma Pacific plate palaeomagnetic pole determined from a skewness analysis of crossings of  
marine magnetic anomaly 25r},  
author={Petronotis, Katerina E and Gordon, Richard G and Acton, Gary D},  
journal={Geophysical Journal International},  
volume={118},  
number={3},  
pages={529--554},  
year={1994},  
publisher={Blackwell Publishing Ltd Oxford, UK}  
}
```

```
@article{woodworth2018paleolatitude,  
title={Paleolatitude of the Hawaiian hot spot since 48 Ma: Evidence for a mid-Cenozoic true polar  
stillstand followed by late Cenozoic true polar wander coincident with Northern Hemisphere glaciation},  
author={Woodworth, Daniel and Gordon, Richard G},  
journal={Geophysical Research Letters},  
volume={45},  
number={21},
```

pages={11--632},
year={2018},
publisher={Wiley Online Library}
}

Reviewer #2 (Remarks to the Author):

The manuscript by Mitchell et al describes a paleomagnetic study of Italian limestones that shows a shift of inclination and declination between about 84 - 78 Ma that is interpreted as an oscillation of true polar wander. The paleomagnetic study itself appears to be very thorough and of high quality. The results appear to be solid. In my previous review, I said that I think the manuscript is significant because it finds a clear signal for small scale TPW "hiding in plain sight" (as it were) in a period of time in which many thought there was no such TPW event. The reason that this study can delineate it is because of the large number of samples and high time resolution. I think that the findings and the discussion are appropriate and important.

I read the rebuttal to reviews and it appears that the authors have worked hard to resolve all issues. I accept that they have. I think the manuscript should be accepted.

I have only a few quibbles about the manuscript. On line 109, I find the wording "single-domain (possibly large) biogenic magnetite grains" to be hard to decipher. Single-domain and biogenic magnetite grains are usually quite small. But it is not clear which is modified by "possibly large" and what that is supposed to mean.

In Figure 2 there are some changes needed in the caption. The plot on the left shows inclination, but for C33r, the inclinations must be absolute values because they don't change sign. So technically, the caption should state that for C33r, positive values were plotted. Also, in the figures, there are some cryptic items. The left panel says "Furlo $f=0.85$ " and the right panel says "Furlo 12°" and "Aprio 5°". These annotations are cryptic and should be explained in the caption.

Otherwise the manuscript is well written and is ready for acceptance.

Best regards,
Will Sager

Reviewer #3 (Remarks to the Author):

I am still not convinced about the claimed TPW oscillations but the authors have done a good job in their rebuttal to the comments and only future studies can prove them right or wrong.

Responses in blue.

Revisions in red.

All line numbers refer to the revised manuscript *without* tracked changes.

Reviewer #1 (Remarks to the Author):

I think the authors have done a solid job of responding to the reviews, in particular to my prior comments. I do have one remaining comment that I didn't bring up in my prior review, but should have, that relates to lines 192-193 and elsewhere. Although global appraisals have not identified TPW events in the past 100 Ma, my group has identified a couple of events mainly from Pacific plate data (but we think they are recorded in the global data, but with much lower precision and resolution). First, we identified a shift of about 8 degrees in Eocene time (Petronotis, Gordon, and Acton, 1994, full citation below), which we hypothesized was caused by TPW. We have lots of recent abstracts strongly supporting that early inference (and a manuscript in revision for Nature Communications).

More recently, Woodworth and Gordon (2018) proposed that about 3 degrees of TPW has occurred since 12 Ma and is probably continuing to occur.

These events are smaller in amplitude than the Cretaceous event investigated by the authors, but the context of the manuscript is incomplete without mentioning them I think.

Revision: We thank the reviewer for making this distinction and offering these references. Both have been added to the most relevant sentence (lines 204-207): "Although TPW for the past 100 Myr may be of smaller amplitude^{1,2} than earlier TPW shifts³, our high-resolution palaeomagnetic study would suggest that one cannot rule out the possibility of rapid TPW shifts of small amplitude (<20°) in the most recent past."

A few other comments and suggestions:

Line 16: Geodesists use the term "contemporaneous" to mean "now" or the past few years or decades. Better to use the term "coeval" I think.

Revision: Good point. We now use the word "simultaneous" instead, which actually reads more smoothly.

Line 24: please insert "non-hydrostatic" in front of "inertia"

Revision: Good point. Term has been inserted.

Line 39: Please clarify that “questionable” applies to testing for true polar wander , but not to reliability for reversal studies. Or, if you mean that too, be clear about it.

Revision: Good distinction. The sentence in question now clarifies our meaning and reads (line 42): “But the reliability of those old (however seminal) data^{4,5} are questionable for directional studies⁶.”

References mentioned above:

```
@article{petronotis199457,  
title={A 57 Ma Pacific plate palaeomagnetic pole determined from a skewness analysis of  
crossings of marine magnetic anomaly 25r},  
author={Petronotis, Katerina E and Gordon, Richard G and Acton, Gary D},  
journal={Geophysical Journal International},  
volume={118},  
number={3},  
pages={529--554},  
year={1994},  
publisher={Blackwell Publishing Ltd Oxford, UK}  
}
```

```
@article{woodworth2018paleolatitude,  
title={Paleolatitude of the Hawaiian hot spot since 48 Ma: Evidence for a mid-Cenozoic true  
polar stillstand followed by late Cenozoic true polar wander coincident with Northern  
Hemisphere glaciation},  
author={Woodworth, Daniel and Gordon, Richard G},  
journal={Geophysical Research Letters},  
volume={45},  
number={21},  
pages={11--632},  
year={2018},  
publisher={Wiley Online Library}  
}
```

Reviewer #2 (Remarks to the Author):

The manuscript by Mitchell et al describes a paleomagnetic study of Italian limestones that shows a shift of inclination and declination between about 84 - 78 Ma that is interpreted as an oscillation of true polar wander. The paleomagnetic study itself appears to be very thorough and of high quality. The results appear to be solid. In my previous review, I said that I think the manuscript is significant because it finds a clear signal for small scale TPW "hiding in plain sight" (as it were) in a period of time in which many thought there was no such TPW event. The reason that this study can delineate it is because of the large number of samples and high time resolution. I think that the findings and the discussion are appropriate and important.

I read the rebuttal to reviews and it appears that the authors have worked hard to resolve all issues. I accept that they have. I think the manuscript should be accepted.

We are very pleased that the reviewer thinks our revised manuscript resolved all issues and is acceptable for publication.

I have only a few quibbles about the manuscript. On line 109, I find the wording "single-domain (possibly large) biogenic magnetite grains" to be hard to decipher. Single-domain and biogenic magnetite grains are usually quite small. But it is not clear which is modified by "possibly large" and what that is supposed mean.

Revision: It is possible that the magnetofossils are large for magnetofossils (e.g., "gigantism" at the PETM, and single-domain behavior is still exhibited despite the exceptionally large crystal sizes⁷). Although such gigantism is possible in the Scaglia Rossa Limestone, this would be entirely speculative at this point. Therefore, we have opted to delete the parenthetical speculation "(possibly large)", which also avoids the potential confusion that the reviewer encountered.

In Figure 2 there are some changes needed in the caption. The plot on left shows inclination, but for C33r, the inclinations must be absolute values because they don't change sign. So technically, the caption should state that for C33r, positive values were plotted.

Revision: Good catch. So as to avoid confusion, the following has been added to the Figure 2 caption (lines 447-448): "reversed polarity data were reversed for mean calculations."

Also, in the figures, there are some cryptic items. The left panel says "Furlo $f=0.85$ " and the right panel says "Furlo 12° and "Aprio 5° ". These annotations are cryptic and should be explained in the caption.

Revision: Good catch. These numbers are now each indicated in the Figure 2 caption.

Otherwise the manuscript is well written and is ready for acceptance.

Best regards,
Will Sager

Reviewer #3 (Remarks to the Author):

I am still not convinced about the claimed TPW oscillations but the authors have done a good job in their rebuttal to the comments and only future studies can prove them right or wrong.

We very pleased to hear the reviewer thinks we have done a good job revising the manuscript.

- 1 Woodworth, D. & Gordon, R. G. Paleolatitude of the Hawaiian hot spot since 48 Ma: Evidence for a mid-Cenozoic true polar stillstand followed by late Cenozoic true polar wander coincident with Northern Hemisphere glaciation. *Geophysical Research Letters* **45**, 11,632-611,640 (2018).
- 2 Petronotis, K. E., Gordon, R. G. & Acton, G. D. A 57 Ma Pacific plate palaeomagnetic pole determined from a skewness analysis of crossings of marine magnetic anomaly 25r. *Geophysical Journal International* **118**, 529-554 (1994).
- 3 Mitchell, R. N. True polar wander and supercontinent cycles: Implications for lithospheric elasticity and the triaxial Earth. *American Journal of Science* **314**, 966-979 (2014).
- 4 Lowrie, W. & Alvarez, W. Late Cretaceous geomagnetic polarity sequence: detailed rock and palaeomagnetic studies of the Scaglia Rossa limestone at Gubbio, Italy. *The Geophysical journal of the Royal Astronomical Society* **51**, 561-581 (1977).
- 5 Alvarez, W. & Lowrie, W. Upper Cretaceous palaeomagnetic stratigraphy at Moria (Umbrian Apennines, Italy); verification of the Gubbio section. *Geophysical Journal of the Royal Astronomical Society* **55**, 1-17 (1978).
- 6 Sager, W. W. & Koppers, A. A. P. Late Cretaceous true polar wander: Not so fast (response). *Science* **288**, 2283a (2000).
- 7 Schummann, D. *et al.* Gigantism in unique biogenic magnetite at the Paleocene–Eocene Thermal Maximum. *Proceedings of the National Academy of Sciences* **105**, 17648-17653 (2008).